# Pseudorabies Virus: From Pathogenesis to Prevention Strategies

**DOI:** 10.3390/v14081638

**Published:** 2022-07-27

**Authors:** Hui-Hua Zheng, Peng-Fei Fu, Hong-Ying Chen, Zhen-Ya Wang

**Affiliations:** 1Zhengzhou Major Pig Disease Prevention and Control Laboratory, College of Veterinary Medicine, Henan Agricultural University, Zhengdong New District Longzi Lake 15#, Zhengzhou 450046, China; zhenghh112@163.com (H.-H.Z.); fpfwdm@126.com (P.-F.F.); 2College of Life Science and Engineering, Henan University of Urban Construction, Pingdingshan 467044, China; 3Key Laboratory of “Runliang” Antiviral Medicines Research and Development, Institute of Drug Discovery & Development, Zhengzhou University, Zhengzhou 450001, China

**Keywords:** pseudorabies virus, pathogenesis, infection, prevention and control

## Abstract

Pseudorabies (PR), also called Aujeszky’s disease (AD), is a highly infectious viral disease which is caused by pseudorabies virus (PRV). It has been nearly 200 years since the first PR case occurred. Currently, the virus can infect human beings and various mammals, including pigs, sheep, dogs, rabbits, rodents, cattle and cats, and among them, pigs are the only natural host of PRV infection. PRV is characterized by reproductive failure in pregnant sows, nervous disorders in newborn piglets, and respiratory distress in growing pigs, resulting in serious economic losses to the pig industry worldwide. Due to the extensive application of the attenuated vaccine containing the Bartha-K61 strain, PR was well controlled. With the variation of PRV strain, PR re-emerged and rapidly spread in some countries, especially China. Although researchers have been committed to the design of diagnostic methods and the development of vaccines in recent years, PR is still an important infectious disease and is widely prevalent in the global pig industry. In this review, we introduce the structural composition and life cycle of PRV virions and then discuss the latest findings on PRV pathogenesis, following the molecular characteristic of PRV and the summary of existing diagnosis methods. Subsequently, we also focus on the latest clinical progress in the prevention and control of PRV infection via the development of vaccines, traditional herbal medicines and novel small RNAs. Lastly, we provide an outlook on PRV eradication.

## 1. Introduction

Pseudorabies (PR), as known as Aujeszky’s disease, was first described in America as early as 1813 and has spread nearly globally since the early 1980s [1]. Its etiological agent is pseudorabies virus (PRV), which has a wide range of hosts; among them, pigs are the natural host and reservoir of the virus. It displays different symptoms at distinct growth phases after being infected with PRV, including the reproductive failure of sows, fatal encephalitis and 100% mortality of newborn pigs, and respiratory distress and growth block of young pigs [2]. For other susceptible animals (ruminants, carnivores and rodents), PRV generally ends with death [3]. In addition, PRV infection might cause endophthalmitis and encephalitis in human beings. Some studies determined the specific sequences of PRV in the patients’ tissues using metagenomic next-generation sequencing, and a human-originated PRV strain hSD-1/2019 was isolated from the cerebrospinal fluid of a patient with acute encephalitis [4,5,6,7,8,9]. Over the years, PRV infection has brought a huge economic loss to the pig industry worldwide and is a serious threat to the health of humans. Although the disease was transiently controlled globally as the result of the use of the glycoprotein E (gE)-negative vaccine Bartha-K61 from Hungary in 1961, PR re-emerged and rapidly spread with a variation of PRV, and the traditional vaccine only offers partial protection against the variant stains [10,11]. Several studies suggest that the PRV variant strains were more virulent to animals and humans than the classical strains [3,5,12]. Moreover, PRV can build a lifelong latent infection in the host’s peripheral nervous system, and infected pigs can potentially be a source of reinfections once the latent viral genome is reactivated. The infection characteristics of PRV variant strains have led to the fact that PR is once again circulating in almost the whole world. It raised scientists’ awareness of the serious threat posed by PRV and encouraged researchers to develop effective interventions.

PRV is an enveloped, linear double-stranded DNA herpes virus belonging to the *Varicellovirus* genus of subfamily *Alphaherpesvirinae* in the family *Herpesviridae* [13]. Its infection generally starts by viral replication in the epithelial cells of the nasal and oropharyngeal mucosa and then spreads to the peripheral nervous system neurons innervating the infected epithelium. Viral particles travel via retrograde transport to the sensory and autonomic peripheral ganglia, where a latent lifelong infection is established [14,15,16]. Upon reactivation, viral replication occurs, and particles spread in the anterograde direction along the sensory nerves back to the mucosal surfaces where the infection initiated. This makes adult pigs and piglets typically exhibit symptoms of respiratory disease and acute neurological disease, respectively [17]. Additionally, PRV infection can also spread via a cell-associated viremia in peripheral blood mononuclear cells from the primary replication site to target organs such as the pregnant uterus, and then secondary replication ensues in the endothelial cells of the pregnant uterus, which can result in vasculitis and multifocal thrombosis, usually leading to abortion [18,19].

Currently, there are no effective means for eliminating PR in the pig population, so the diagnosis, prevention and control of PR are particularly important for the pig industry. In this review, we briefly describe the structure of PRV virions and the viral life cycle. Then, we discuss recent advances in understanding the pathogenesis of PRV infection and its molecular characteristics. Subsequently, the current PRV diagnosis methods are summed up, and we also highlight the latest progress in the prevention and control of PRV infection, including the development of vaccines, Chinese herbal medicines and novel small RNAs. Finally, we look forward to the prospects of PRV eradication in the future.

## 2. The Virion Structure, Genome Structure and the Life Cycle

### 2.1. The Virion Structure

PRV mainly contains two subtypes (I and II). Similarly to other Herpes virions, PRV virions (Figure 1) are approximately 225 nm in diameter and consist of four morphologically distinct structural components, including a linear double-stranded DNA genome, an icosahedral protein capsid, a protein tegument layer, and a lipid envelope containing viral glycoproteins [2,20,21,22]. The double-stranded DNA genome of ~145 kb in length, which can encode more than 70 proteins, is encapsulated in an icosahedral capsid. The tegument is a collection of approximately 12 proteins organized into at least two layers, one of which interacts with envelope proteins, while the other is closely associated with the capsid. The envelope is a lipid bilayer infused with transmembrane proteins, many of which are modified by glycosylation.

### 2.2. Genome, Gene Content and Role in Viral Replication

*Alphaherpesvirus* genomes have a partial colinear arrangement of genes encoding similar functions. Based on the overall arrangement of repeat sequences and unique regions, the herpesvirus genomes can be divided into six classes, designated by the letters A to F [23]. The PRV genome belongs to the D class and is a linear, double-stranded and sense viral DNA genome of approximately 145 kb, with a 74% content of G + C. It is also characterized by two unique regions, which are the unique long region (UL) and unique short region (US), and the US region is flanked by the internal and terminal repeat sequences (IRS and TRS, respectively) of 15 kb in length (Figure 2) [2]. The sequence and gene arrangement of the entire PRV genome are known, and a map of the likely transcript organization, well supported by experimental data, has been established. Recombination between the inverted repeats can produce two possible isomers of the genome, with the US region in opposite orientation, and two isomers are both infectious. PRV has three origins of replication, with one of *OriL* located in the UL region and the others (*OriS*) located in the inverted repeats [24,25]. In addition, the full-length genome of PRV contains 73 different genes encoding a total of 70~100 proteins, which are mainly the capsid proteins, envelope proteins, tegument proteins and various enzymes. Half of them are nonessential proteins for PRV replication [2]. The PRV genes can be divided into three types based on their different functions: structural genes, virulence genes and regulatory genes. They can also be divided into immediate-early genes, early genes and late genes according to the transcription sequence of PRV invading host cells [26].

Some major genes are associated with the process of PRV infection. PRV thymidine kinase (TK), namely the UL23 gene, plays a decisive role in the virulence of the virus. It was primarily described as being related to the replication and neuro-invasiveness of PRV in the central nervous system and is also involved in re-activating the virus during the latent infection period [27]. The lack of the TK gene significantly reduces the ability of replication and transmission in nerve cells without affecting its immunogenicity [28]. As one of the virulence genes, gE, located in the US region, is not essential for viral replication and has no effect on viral immunogenicity. gE glycoprotein can promote the fusion of PRV and cells and mediate the spread of the virus between cells. PRV without the gE gene can only infect the primary trigeminal nerve and sympathetic nerve regulating nasal mucosa, but not the secondary ganglion and sympathetic neurons [29]. In the processes of PRV invading and spreading into the nervous system, gE and gI proteins generally exist in a complex, which is often distributed in the cell membrane of infected cells and in the virus envelope [30]. gI protein is a membrane protein that can not only promote the secretion of gE protein in the endoplasmic reticulum to ensure its correct glycosylation, but also facilitate the transmission of the virus between cells [31]. Furthermore, gI glycoprotein can also cooperate with gC protein in the release process of the virus. The gC protein plays a role in the first step of PRV replication, that is, the adhesion process to cells. It can initiate virus attachment to cells by binding to heparan sulfate (HS) proteoglycans and also participate in the process of virus release from cells [32,33]. The sequence of the gB gene is more conservative than other genes, and gB glycoprotein is an important structural protein of the virus envelope. Both gB glycoprotein and the gH/gL complex can jointly promote the fusion of cell membrane and virus envelope when the virus invades cells [34]. gB protein is a typical class III post-fusion trimer that binds membranes via its fusion loops (FLs) in a cholesterol-dependent manner [35]. gD glycoprotein accelerates the rapid fusion between the PRV capsule membrane and the cytoplasmic membrane of target cells and boosts the fusion of gB/gH/gI complex [36]. Yet, it cannot participate in virus spread between cells, which is not essential for viral replication. Compared with herpes simplex virus (HSV), the gD protein of PRV can only bind to the cell surface receptor Nectin-1 at sites N77, I80, M85 and F129, with a binding ratio of 1:1, and the binding site F129 plays an important role in invading cells [37]. The gG gene locates in the US region. The gG protein belongs to a larger complex, which is synthesized, secreted and released by infected cells, but virus particles do not contain this glycoprotein [36]. Furthermore, gG protein has very good immunogenicity and can effectively stimulate the organism to produce antibodies [38]. At the same time, gG protein can bind to chemokines produced by the organism, leading to the immune escape of the virus. IE180 gene encodes 1460 amino acids with a protein of ~153 kDa in molecular mass, and it is the only immediate early gene in PRV that can be transcribed independently. The accumulation of IE180 protein can start the transcription of other genes, and it is highly similar to some regions of herpes simplex virus type I ICP4 protein, which is complementary to both proteins in some of their functions [39]. The IE180 3′UTR end sequence can form a G-quadruplex structure and inhibit gene transcription [39]. The G-quadruplex ligand small molecule TmPyP4 (meso-Tetrakis (N-methyl-4-pyridiniumyl) porphyrin) can stabilize this structure and further inhibit the early proliferation of PRV [40]. Additionally, IE180 gene transcription locates at the beginning of the PRV replication cycle, which is of great significance for PRV replication, suggesting that the IE180 gene may also become an important target for the development of anti-PRV drugs. As one of the non-essential genes for virus replication, the EP0 gene, located in the UL region, can encode 1230 amino acids and has an early protein of about 45 ku. It can transactivate the immediate early gene IE180. The replication ability of PRV without the EP0 gene in cells is weakened, but it does not affect the virulence. EP0 protein can interact with IE180 protein to activate the transcription of the TK and gG genes; besides, it is located in the nucleus, and the nuclear entry of EP0 protein is co-regulated by Ran protein and input proteins α1, α3 and β1 [41]. The UL21 gene, located in the UL region, is a non-essential gene for PRV replication, and the protein encoded by UL21 belongs to the tegument proteins. The proliferation ability of PRV without the UL21 gene is reduced, but can be restored on pUL21 compensatory cells [42]. Exogenous pUL21 can inhibit the NF-kB pathway, with a positive correlation, and its carboxyl end can interact with cytoplasmic dynamic protein Roadblock-1, which further influences the nerve infectivity of PRV [42]. The UL41 gene is an essential gene for virus replication and encodes host closure protein (named as vhs), with a molecular weight of about 40 ku. The vhs protein has ribonuclease activity both in vivo and in vitro, and can degrade the mRNA of host cells and inhibit gene expression. The vhs protein can also cleave the downstream region of internal ribosome entry site (IRES), and the translation initiation factors eIF4H and eIF4B can significantly increase the RNase activity of recombinant PRV vhs against capped RNA [43,44].

### 2.3. The Life Cycle of PRV

The process by which PRV virions enter host cells is primarily initiated by the binding of virions to the surface molecules of host cells and the fusion of the virus and host cell membranes (Figure 3). PRV virions first attach to cells by the interaction of gC with heparan sulfate proteoglycans in the extracellular matrix. PRV gD then binds to specific cellular receptors to stabilize the virion-cell interaction. Finally, PRV gB, gH and gL mediate the fusion of the viral envelope and the cellular plasma membrane to allow penetration of the viral capsid and tegument into the cell cytoplasm, and tegument proteins in the outer layer quickly dissociate from the capsid following their fusion [45,46]. Then, the capsid interacts with dynein for transport along microtubules from the cell periphery to the nuclear pore [45,47]. After capsid docking at the nuclear pore, the PRV DNA is released into the nucleus from intact capsids [45,47].

For the viral transcription and cascade, the immediate-early IE180 transcript is detected within 40 min of infection, and its protein is synthesized up until 2.5 h post-infection (hpi). The IE180 protein regulates the early gene expression related to replication. The early EP0 transcript is detected at 2 hpi, and its expression in vivo activates gene expression from PRV promoters, such as IE180, TK and gG, which have the characteristics of transcription activators. Other regulators of gene expression also participate in viral transcription, including UL54, UL41, and UL48, and among them, UL54 and UL41 are likely to encode potent regulators of both viral and cellular gene expression. UL41 encodes the vhs protein, which is responsible for the virion host shut-off of cellular protein synthesis. UL48 encodes the tegument protein VP16, which enhances the expression of viral immediate-early genes in newly infected host cells [48].

Upon entry into the host nucleus, the linear viral DNA genomes assume a circular form and are quickly repaired of nicks and misincorporated deoxyribonucleotides [2]. The circular genomes serve as the template for DNA synthesis, and the initial theta replication mechanism quickly switches towards a rolling-circle mechanism of DNA replication. The latter process produces replicated DNA in the form of long linear concatemeric genomes that serve as the substrate for genome encapsidation [49,50]. In this process, many of the enzymes encoded by PRV also participate in viral DNA replication, such as UL52, UL42, UL30, UL29, UL9, UL8, and UL5. Several enzymes encoded by PRV genomes can be involved in nucleotide metabolism, for example, dUTPase (UL50), thymidine kinase (UL23) and two-subunit ribonucleotide reductase (UL39/UL40). The viral genome also encodes a uracil DNA glycosylase (UL2) as well as an alkaline nuclease (UL12), which both serve in viral DNA repair, recombination and DNA concatemer resolution [27,51,52]. When the viral genome completes transcription, translation and DNA replication, the capsid protein automatically enters the nucleus of the cell to form the basic assembly unit of the capsid and assembles into the nucleocapsid in the nucleus of the cell. After assembly, PRV nucleocapsids cross the inner and outer nuclear membrane by budding and fusion, respectively, following release into the cytoplasm, and then are formed into the mature enveloped virions and released to the outer cell by budding for the next round of infection [53,54,55].

## 3. Occurrence and Development of PRV Infection

PRV can cause respiratory disease, neurological disorders and abortion in pigs. Its transmission mainly occurs through direct contact between oral and nasal secretions but can also occur by aerosols, transplacental contact and blood [56,57]. In this work, we reviewed the main steps in the pathogenesis of PRV in pigs (Figure 4).

### 3.1. PRV Primary Replication in the Upper Respiratory Tract

After PRV enters a natural host, it firstly replicates by an infection foci manner in the epithelial cells lining the upper respiratory tract (URT), including nasal septa, tonsils, nasopharynxes, trachea and lungs [14,17,58,59]. In vivo, viral DNA was detected in the nasal mucosa, tonsils and lungs of 2-week-old piglets starting 24 hpi, and PRV-induced plaques can be observed in the epithelium of porcine nasal mucosa explants after 24 hpi in ex vivo experiments [17,60,61]. Primary PRV infection in multiple tissues of porcine URT causes the destruction and erosion of epithelium, with slight respiratory symptoms consisting of sneezing, coughing, dyspnea and nasal discharge after 3 to 6 days post-inoculation (dpi), which normally disappear quickly [59,62]. Besides, viral shedding can be detected in nasal secretions starting from 1 to 14 dpi [63].

### 3.2. PRV Replication in the Draining Lymph Nodes and Viremia

After respiratory epithelium infection, PRV can pass the basement membrane (BM) by infected leukocytes to penetrate the connective tissues and further reach the bloodstream and the draining lymph nodes [61,64]. The process of viral invasion into lamina propria through BM is mediated by the activity of trypsin-like serine protease excreted by the virus [65]. PRV antigens or DNA are both detected in the inguinal lymph nodes of pigs starting from 24 hpi to 48 hpi, and the virus can persist for 35 days in pharyngeal lymph nodes [66,67,68,69]. PRV infection is also amplified in the draining lymph nodes, and infected leukocytes are discharged into the blood circulation through the efferent lymph. Hence, PRV induces cell-associated viremia in peripheral blood monocytes and promotes its transmission in pigs [18,70]. Meanwhile, viremia also occurs in cell-free form after PRV infection and can be regularly detected between 1 dpi and 14 dpi [18]. Cell-associated viremia is considered a prerequisite for the dissemination of PRV to the pregnant uterus in pigs.

### 3.3. PRV Entry into the Peripheral Nervous System (PNS) Neurons and Spread to the Central Nervous System (CNS)

Upon primary replication of respiratory epithelium in adult pigs after PRV primary infection, it enters PNS’ nerve endings, which contain those coming from the sensory trigeminal ganglia (TG) and olfactory bulb, and other facial, parasympathetic, sympathetic nerve neurons that innervate the epithelium [71,72]. PRV particles can be transported retrogradely to sensory and autonomic peripheral ganglia. Generally, the herpesviruses are able to establish a reactivable, latent infection in their hosts. PRV can also set up a lifelong potential infection in PNS neurons of pigs, whereas the infected pigs cannot display any clinical symptoms after recovering from the respiratory disease [15,16,63,73,74]. After stress-induced reactivation, viral replication happens in the PNS ganglia, and virions spread in the anterograde along the nerves to the mucosal surfaces where the infection initiated, further causing mild respiratory signs in adult pigs upon viral reactivation [75]. In addition, PRV barely propagates to the CNS in the retrograde direction to result in the encephalitis of adult pigs, but its latent period and reactive cycles in pigs cause the infectious virus to fall off and spread to uninfected pigs, which is conducive to viral accumulation in pig populations. Intriguingly, the herpesviruses of human beings and other animals, such as varicella-zoster virus (VZV) and bovine herpesvirus type 1 (BHV-1) have a similar way of spreading to invade PNS neurons [76,77].

### 3.4. Secondary Replication in the Swine Pregnant Uterus

Once in the blood circulation, PRV-infected monocytes can cross the endothelial cells (EC) barrier of the maternal blood vessels to reach the pregnant uterus of sows via adhesion and fusion of these monocytes with EC, further transmitting PRV [78]. The adhesion molecules on the surface of EC and leukocytes play a significant role in the infection of the vascular endothelium, and the secondary replication in the EC of the pregnant uterus can lead to vasculitis and multifocal thrombosis, with an abortion of a sow [18,79,80]. The EC infection in the vasculature of the pregnant uterus is usually mediated by intercellular contact between infected monocytes and EC [78]. The occurrence of abortion may depend on the hormonal activity and immune status of sows during pregnancy. In fact, it has been proved that the expression of adhesion molecules on EC is induced by cytokines and hormones in the local environment during pregnancy [81,82,83]. These cytokines may accelerate the adhesion of infected monocytes to the endothelial cells.

Upon the intranasal, intra-uterine, and intra-fetal inoculations of vaccinated pregnant sows, PRV antigen can be detected in vaginal and sacral ganglia [84]. An extensive EC infection can lead to the fetal membranes shedding in early pregnancy, resulting in a virus-negative fetal abortion or fetal reabsorption in sows. Little uterine vascular pathology may cause transplacental infection and the abortion of virus-positive fetuses in the second and third trimester of gestation or a stillborn pig [59,85]. A fetus with viral abortion generally displays several lesions, such as necrosis of the liver, spleen and lungs, and PRV strains can be isolated from the above organs [19,86].

### 3.5. PRV Infection in Suckling and Weaned Piglets

PRV infection usually brings more severe lethality to piglets than to adult swine and sows [59]. In newborn piglets, sudden death usually occurs in the absence of clinical signs. Instead, before the death of suckling pigs, some signs are found in infected pigs, including fever, vomiting and CNS symptoms, which consist of coordination problems, hindquarter weakness, convulsions and paralysis. It is worth noting that the mortality rate of newborn and suckling pigs is close to 100%. In weaner pigs, clinical signs are similar to those of suckling pigs, with a mortality rate of 5% to 10%. Nevertheless, pigs of any age cannot have itching. Infectious viruses can be isolated from brain tissue samples of piglets naturally infected with PRV [64]. The severity of symptoms diminishes with age, as adult pigs have a more effective immunity than piglets.

### 3.6. PRV Infection in Humans

In the past century, the viewpoint of PRV infecting humans has been controversial due to the absence of unequivocal etiology or serological diagnosis [87]. The first case of humans with suspected PRV infections was reported in 1914, and they were previously in contact with PRV-infected cats. Subsequently, several patients with PRV infection were reported after long-term exposure to PRV-susceptible animals (pigs, cats, dogs). These patients mainly displayed pruritus, headache, fever, swelling, sweating, dysphagia, aphthous ulcer, altered mental status, seizure and coma [4,5,6,7,8]. Notably, Chinese patients mainly showed encephalitis and endophthalmitis, and most of them were working on pig farms and had injuries to their fingers or other places on their bodies. In addition, the first human-originated PRV strain, designated hSD-1/2019, was isolated and identified from the cerebrospinal fluid (CSF) of PRV-infected patients worldwide, which provided direct evidence of PRV infection in human beings [5]. In terms of the PRV infectious pathway in humans, it has yet been unclear. However, from the perspective of animal research, it was supposed that PRV might not only affect the brain, but also affect other human organ systems, further causing serious consequences. Although these patients all survived until hospital discharge, and clinical symptoms disappeared, it was unclear whether the recovered patients still carry PRV owing to the long-term latent period of PRV in pigs. Therefore, it is necessary to ensure thorough skin protection in humans, particularly those who have been in close contact with pigs.

## 4. Genetic Evolution of PRV

Currently, PRV strains in the world can be divided into two genotypes, I and II. The genotype I strains are mainly prevalent in Europe, America and parts of China, while the majority of genotype II strains are isolated from Asia, particularly China [88]. The latter has undergone mutations caused by host immune pressure over a long period of time and evolved into novel PRV variant strains, which then led to the co-prevalence of both variant and classical strains, which again poses great threats to the swine industry in China [89]. Because gB, gC and gD glycoproteins are major immune-related proteins and gE is a significant virulence protein, these four gene sequences are often used to analyse PRV’s genetic evolution [90,91]. To investigate the genetic characteristics of PRV strains from various countries and species, phylogenetic trees based on the complete length of gE, gC, gB and gD gene sequences were constructed using MEGA software (version 7.0) by the neighbor-joining method with 1000 bootstrap replicates [92]. The results showed that these PRV strains were grouped into two genotypes as expected (Figure 5). The genotype I group is composed of European–American PRV strains, but the genotype II group is formed by Asiatic strains (China and Japan), and among them, Chinese PRV variant strains were clustered in one subgroup. Findings revealed that PRV genotype II strains had become dominant prevalent strains instead of genotype I in China, which is in agreement with previous studies [88,90,93]. This finding again explains why Bartha strains (genotype I) did not provide full protection against variant PRV strains (genotype II). In addition, PRV strains isolated from other animals are randomly distributed in two genotypes (Figure 5). Combined with previous research [94], this suggests that PRV strains isolated from animals and human beings may have a similar ancestor to those of pigs.

In mutation analysis of PRV variant strains, He et al. found that the average amino acid (aa) differences of the Qihe547 variant strain are 4.94%, 1.16% and 0.46% compared with PRV strains of genotype I and the classical and variant strains of genotype II, respectively [88]. Compared with genotype I strains, PRV genotype II strains present high genetic mutations in internal and terminal repeat regions, and nucleotide insertions, deletions and mutations are commonly observed in the different PRV genes [95]. For example, there are two insertions of discontinuous six nucleotides at sites 142–144 (GAC) and 1488–1490 (CGA) in the gE gene of PRV strains in genotype II [96,97]. In Sun’s study, PRV strains in genotype II are found to have three aa deletions (75VPG77) in the gB protein and seven aa insertions (63AASTPAA69) in the gC protein [98]. Interestingly, two aa deletions (288SP289) have been identified in the gD protein of Chinese PRV variant strains by comparison with classical Chinese strains [99]. Furthermore, inter-clade and intra-clade recombinant events can accelerate the evolution of the PRV genome and alter viral virulence, which has been demonstrated in several studies [88,90,100]. In Zhai’s study, two inter-clade recombinant PRV strains (FJ-W2 and FJ-ZXF) were reported, and their gE, gC and gD genes were assigned to genotype II, whereas gB genes belong to genotype I [90]. Subsequently, Huang et al. analyzed the genetic evolution of the primary immune-related gene sequences of PRV variants, and the results showed that the gB gene of the PRV variant strain FJ62 isolated from piglets in Sichuan, China, is identical (100%) to the MY-1 strain (No. AP018925) from a wild boar in Japan, with low sequence homologies (98.4–98.5%) of Chinese PRV strains. However, its gC, gD and gE genes have high sequence similarities of 99.5%, 99.9%, and 99.9%, respectively, demonstrating that PRV variant strain FJ62 may appear from a recombinant event of PRV strains of genotypes I (Japan) and II (China) spanning different countries [91]. In another report, PRV HLJ-2013 was isolated from pigs in Heilongjiang province of China belonging to genotype II, and its genome sequences are derived from three viruses (including a yet unknown parental virus, the European viruses and the same ancestor of all Chinese strains) based on the phylogenetic trees of both protein-coding genes and non-coding regions [100]. The recombination analysis showed that there are six recombinant events in SC strains (No. KT809429) belonging to genotype II, and among these events, HLJ-2013 is predicted to be the major parent of the SC strain, with a minor parent of Bartha [100]. Additionally, Tan et al. also found that a naturally recombinant event might occur in the genome of the HN-2019 strain isolated from a sick piglet in Hunan of China, between the PRV classical strain and the HB-98 vaccine strain, which again confirmed the presence of a recombinant event in PRV [101].

## 5. Diagnostic Methods

At present, serological technologies and molecular biology methods have become the common diagnostic approaches for PRV detection (Figure 6) because traditional clinical and pathological diagnostic methods cannot accurately diagnose PR. Two common methods are used to verify the PRV infection based on PRV-specific antibodies and nucleic acids, respectively, which have their own features (Table 1).

### 5.1. Serological Approaches for the Detection of PRV Antibodies

Due to the extensive use of PRV gE-deleted vaccines worldwide, gE as the marker antigen has become widely used in serological approaches, and many PRV gE antibodies have been developed to quickly and effectively differentiate infected from vaccinated animals (DIVA). gB antibodies based on serological methods can also be used for monitoring the immune level induced by vaccine immunization. Up to date, various serological methods can be used for the detection of PRV antibodies, such as the direct-immunofluorescence method (DFM), indirect immunofluorescence assay (IFA), serum neutralization test (SNT), enzyme-linked immunosorbent assays (ELISA), blocking immunoperoxidase monolayer assay (b-IPMA), latex agglutination test, agar diffusion test, particle concentration fluorescence immunoassay (PCFIA) and immunochromatographic strip [1,93,101,102,103,104,105,106,107,108,109]. Among them, ELISA remains the most common method in the clinical detection of PRV antibodies because it has high specificity and sensitivity compared with other screening assays. There are three single commercial serum antibody ELISAs, including the gB blocking ELISA (gB bELISA), gI blocking ELISA (gI bELISA) and gE indirect ELISA (gE iELISA) [110]. To make detection more convenient, competitive ELISAs (cELISA) targeting the gB or gE antibody have been developed and extensively applied in China [111,112,113,114]. Interestingly, a novel serological technology based on the blocking fluorescent lateral flow immunoassay takes less time for PRV detection and is sensitive to differentiate wild PRV-infected and vaccinated pigs, whereas a commercial gE-ELISA kit is not [102]. Subsequently, another new detection method, dual fluorescent microsphere immunological assay (FMIA), was developed for detecting PRV gE and gB IgG antibodies simultaneously, and it also has accuracy for gE detection with high sensitivity (92.3%) and specificity (99.26%) compared to a commercial gE/gB ELISA kit, with less time and cost expenses [115]. Furthermore, both the immunochromatographic assay and liquid chip technology methods also exhibit higher sensitivity than that of cELISAs [116]. Hence, blocking fluorescent lateral flow immunoassay, FMIA, immunochromatogragphic assay and liquid chip technology are expected to become new clinical laboratory diagnostic methods for detecting PRV antibodies, although these methods are not universal thus far.

### 5.2. Molecular Biology Approaches for the Detection of PRV Infection

To further improve the sensitivity and specificity of PRV detection, molecular biology methods targeting the specific sequences of PRV genes, including the gE, gI, gC, gD, gB, and gG genes, have been established (Table 2), such as polymerase chain reaction (PCR), real-time PCR (RT-PCR), TaqMan real-time PCR (qPCR), nano PCR, droplet digital PCR (ddPCR), real-time recombinase-aided amplification (RT RAA), loop-mediated isothermal amplification (LAMP), and duplex fluorescence melting curve analysis (FMCA) [117,118,119,120,121,122,123,124,125,126,127,128,129,130,131]. Generally, PCR and RT-PCR are the most frequently used approaches to quickly detect PRV infection or distinguish between the PRV wild-type and vaccine strains [124]. In particular, multiplex PCR assays have also been developed for simultaneously detecting PRV and other pathogens, for example, porcine circovirus 3 (PCV3), porcine circovirus 2 (PCV2), porcine parvovirus (PPV) and porcine cytomegalovirus (PCMV), and Torque teno sus virus 1 and 2 [118,132,133,134,135,136]. In addition, the novel high-throughput sequencing, next- and third-generation sequencing (NGS and TGS) methods are used to survey the transcriptome of PRV, and the existence of PRV can be detected in patients by NGS, which is the most powerful and supersensitive assay [6,137]. Nevertheless, high-throughput sequencing is not suitable for wide-range clinical detection owing to its high cost [6]. Compared with conventional PCR, the LAMP assay is more sensitive, specific, rapid and cost-effective, so it is more suitable for PR diagnosis in the field, with a huge potential application in the prevention and control of PRV [122].

### 5.3. Other Approaches for the Detection of PRV Infection

In addition to the above methods, direct detection of pathogens is another effective candidate, which also greatly facilitates the vaccine design to prevent PR. The methods mainly include virus isolation and subsequent laboratory diagnosis, involving serology, molecular biology and electron microscopy. Virus isolation (VI) is considered the “gold standard” for pathogen diagnosis, and a large number of PRV strains from different hosts have been successfully isolated, though this method is only applicable to professional laboratories [138,139]. Because the traditional diagnostic methods (VI identification and animal experiments), serological diagnostic approaches and molecular biological methods have some limitations, such as complicated operation and high technical requirements, and are time-consuming, a paper biosensor doped with Fe_3_O_4_@SiO_2_–NH_2_ and multi-walled carbon nanotubes (MWCNTs) for the rapid detection of PRV has been developed [140]. In this assay, Fe_3_O_4_@SiO_2_–NH_2_ can provide magnetic response characteristics, and MWCNTs are able to increase the electrical conductivity [140].

## 6. The Prevention of PR

To better prevent and control PR, numerous efforts have been made for the development of effective means to control PRV infection, mainly including vaccines and other novel viral inhibitors.

### 6.1. Main Vaccines against PRV Infection

As a great challenge, PRV has been prevalent in pig farms worldwide for nearly two hundred years. Vaccination is one of the most effective ways to prevent disease and minimize the economic losses caused by PR [10]. Most PRV vaccines are live gene-modified virus vaccines (Table 3). The initial live gene-modified vaccines (attenuated Bartha-K61 strain and PRV Bucharest strain) are usually obtained from extensive passaging of virulent field isolates in cell cultures from 1961, which, following the wide application in pig herds, and effectively controlled PR worldwide [141]. In China, a live gene-modified vaccine (attenuated Bartha-K61 strain) was imported from Hungary in the 1970s and was widely inoculated in pig farms, with effective control of PR [12,142]. With further understanding of PRV genetics and molecular biology, gE protein has become regarded as the important swine neurovirulence factor [143]. TK is also essential for virus replication in nonmitotic tissues (neurons), and other proteins are considered the factors of viral replication, such as gI, Us9, Us2, gC, gG and PK proteins [144]. Therefore, the genes encoding these proteins can be deleted to mediate viral attenuation, especially the gE gene. In fact, TK-negative stains (∆TK) is the first genetically modified live vaccine and was licensed for application in 1985 [27]. Subsequently, other gene deletion vaccines also generated by various technologies, such as gE gene deletion and double-gene gE/TK deletion of the NIA3 strain, double-gene gD/gI deletion of the NIV-3 strain, double-gene TK/gG deletion of the virus strain, triple-gene TK/gE/gI deletion of the BUK strain and Fa strains, four-gene gD/gG/gI/gE deletion of the PrV(376) strain, and a triple-gene-deleted (gE/gI/TK) vaccine generated based on the PRV Fa strain licensed in 2003, which was regarded as the first genetically modified vaccine against PR in China [145,146,147,148,149,150]. However, since 2011, PR outbreaks caused by emerging PRV variants have occurred in Chinese pig herds which were immunized with the Bartha-K61 strain, indicating that the classical attenuated PRV vaccine cannot provide complete protection for swine [111,151,152,153]. With the rapid development of biotechnologies and in-depth understanding of the biological functions of PRV encoding genes, some gene-modified vaccines and other types of vaccines have also been generated based on emerging virulent PRV strains. As ideal target genes, the gE, gI, TK, Us9, Us2, gC, gG and PK genes can be deleted to develop genetically engineered vaccines against re-emerging PR, particularly gE, gI and TK genes. Therefore, similar to single- or multiple-genes deletion of classical PRV strains, some novel gene-modified vaccines based on the variant PRV strains have been generated in recent years, but only two types of vaccines have been licensed thus far, including the gE-gene-deleted inactivated vaccine on the basis of the PRV HeN1201 strain isolated from 2019 and another natural four-gene-deleted (gI/gE/Us9/Us2) vaccine based on the PRV C strain in 2017 [29,154,155]. There are other candidates, including killed and live attenuated vaccines based on the variant PRV strains, and the latter can be generated by homologous DNA recombination, clustered regularly interspaced palindromic repeats (CRISPR)/associated (Cas9) system and bacterial artificial chromosome (BAC) [155,156,157,158]. Thus far, the gene-modified vaccine of PRV based on the variant PRV strains contains a double-gene deletion of the ZJ01 strain (vZJ01-ΔgE/gI), the PRV-XJ strain (rPRVXJ-delgI/gE-EGFP), and the AH02LA strain (PRV B-gD&gC^S^), three-gene deletion based on the HN1201 strain (vPRV HN1201 TK^-^/gE^-^/gI^-^), the TJ strain (rPRVTJ-delgE/gI/TK), the NY strains (rPRV NY-gE^-^/gI^-^/TK^-^), the GX strain (rGXΔTK/gE/gI), the XJ5 strain (rPRV/XJ5-gE^-^/gI^-^/TK^-^) and the ZJ01 strain (rZJ01-ΔTK/gE/gI), four-gene deletion of the PRV-GDFS (PRV GDFS-delgI/gE/US9/US2) and ZJ01 strains (rZJ01-ΔgI/gE/TK/UL13), and five-gene deletion of HN1201 (rHN1201^TK−/*gE*−/*gI*−/11*k*−/28*k*−^), which have been proved to be effective in preventing PR caused by mutant strains [29,155,156,157,158,159,160,161,162,163,164,165]. Considering the safety of these candidate strains in field applications, further clinical trials must be conducted.

Additionally, the large genome of PRV can serve as vaccine vectors for expressing exogenous antigens without affecting its infectivity and immunogenicity [180]. Several live PRV-based vector vaccines encoding significant antigens of other animal pathogens have been generated and represented in the previous overview [181]. Similar to the gene-modified vaccine, gG, gI, gE, and TK genes generally are used to insert exogenous sequences [182,183,184]. At present, varied foreign genes encoding protective antigens of other pathogens have been successfully inserted into the large genome of PRV, including the PPV VP2 gene, classical swine fever virus (CSFV) E2 gene, the Brucella melitensis Bp26 gene, and some genes of the African swine fever virus which include CP204L (p30), CP530R (pp62), E183L (p54), B646L (p72) and EP402R (CD2v) [116,183,184,185,186,187,188]. Generally speaking, most PRV recombinant vaccines have not yet been licensed so far, although these vaccines can effectively prevent multiple infectious diseases simultaneously. Surprisingly, recombinant targeted *Bacillus subtilis* vaccine expressing PRV gC and gD proteins can effectively induce a mucosal immune response against this disease in recent study [189], and efficient protection provided by a gD-based subunit vaccine against PRV variant infection in pig models was also confirmed [190].

For China, many researchers have developed diverse types of anti-PRV vaccines, mainly including inactivated vaccines, live attenuated vaccines and live PRV-based vector vaccines, and most of them can produce high levels of neutralizing and gB antibodies, with effective protection against PRV. Furthermore, these candidate vaccines also follow the distinction between the infected and vaccinated animals (DIVA), which is beneficial to PR eradication in China. Different types of vaccines have different advantages. The inactivated vaccine is highly safe for vaccinated animals without viral virulence reversion, but it is generally less effective than both live vaccines [10]. Notably, two live vaccines display disadvantages, for example, their safety in pigs and non-target animals. Some studies reported that vaccination with a gene-modified PRV strain could lead to PR in sheep with severe clinical signs, as well as adult red foxes, and also pose a potential threat to the health of dogs [191,192,193]. In general, lengthy testing should be conducted on genetically modified live vaccines [167].

### 6.2. Chinese Herbal Medicines as Potential Anti-PRV Drugs

Chinese herbal medicine has a history spanning thousands of years and has been extensively applied in the treatment of various diseases of human beings, such as atherosclerosis, sepsis, diabetes, cancers, chronic kidney disease and anti-SARS-CoV-2 (COVID-19) infection [194,195,196,197,198,199,200]. Some researchers have also found that Chinese herbal medicine can improve some animal diseases [201,202,203,204]. Sinomenine, a comment agent in Chinese herbal medicines, can decrease the incidence and severity of certain LPS-induced toxicities, for example, cell adhesion, systemic inflammation and multiple organ dysfunction [201]. Ginger extracts can relax and vasoprotect porcine coronary arteries [203]. With the international recognition of traditional Chinese herbal medicine in disease treatment, this medicine has gradually gained comprehensive attention. In light of the tremendous impact of PR on the swine industry, some scientists have also committed to looking for inhibitors against PRV infection (Table 4) [205,206,207,208,209]. As a natural phenolic compound, resveratrol (trans-3, 4, 5-trihydroxystilbene; Res) has a variety of properties, such as immunomodulatory, anti-inflammatory and antiviral activities, and especially, its antiviral activities against PRV infection have been fully recognized in numerous examinations [209,210]. Based on these bioactivities, it appears that Res can inhibit the proliferation in PRV-infected piglets and protect rotavirus-infected piglets by reducing the inflammatory response and enhancing immune function [211]. In vitro, Res effectively inhibits PRV replication in a dose-dependent manner, with a 50% inhibition concentration of 17.17 μM, and its inhibition of PRV-induced cell death and gene expression may be related to IκB kinase degradation [211]. The ability of Res anti-PRV and immune-adjuvant was also corroborated in both mice and pigs, and Res could inhibit the replication of ASFV in vitro [142,186,212]. Taken together, these traditional Chinese herbal medicines have great potential value on the inhibitive ability of PRV infection, and they need to be further studied to be promoted to be an effective choice for animals or human beings against PRV challenge.

In addition to Res, other types of compounds with anti-PRV activity have been identified to be effective in vitro, such as kaempferol, panax notoginseng polysaccharides, germacrone, plantago, quercetin, isatis indigotica, radix isatidis, marine *Bacillus* S-12–86 lysozyme, diammonium glycyrrhizin, vanadium-substituted Heteropolytungstate, graphene Oxide, ivermectin and phosphonoformate sodium, whereas some of them cannot be verified for the inhibition of viral replication in vivo (Table 4) [99,213,214,215,216,217,218,219,220,221]. As a novel anti-PRV drug, quercetin is a natural product that has anti-oxidant, anti-bacterial, anti-cancer and anti-viral activities [216,222]. In the latest research, it was found that quercetin can indeed reduce the extent of PRV infection in virus-infected cells in a concentration-dependent manner, suggesting that quercetin mainly holds back the entry of PRV into the host cell by preventing its adsorption to the cell surface [216]. In addition, it is important for a viral infection that quercetin is able to insert into the substrate-binding pocket of PRV gD protein on the PRV surface and connect the N-ring and spiral alpha3 by hydrogen bonding [216]. The anti-PRV activity of quercetin has been demonstrated in mice, indicating that mice inoculated with quercetin can resist the lethal challenge of PRV and decrease the viral loads in the brain [216]. Although quercetin has a powerful therapeutic property against PRV infection both in vitro and vivo, it still needs long-term validation before it can be widely applied in veterinary clinics.

**Table 4 viruses-14-01638-t004:** Overview of compounds with anti-PRV infection activity.

Source	Mechanism	50% Effective Concentration	50% Cytotoxic Concentration	PRV Strain	In Vitro	In Vivo	References
Resveratrol (Res)	The inhibition of viral proliferation, IκB kinase activation	17.17 ± 0.35 μM	Above 262.87 μM	Rong A strain	Yes	Yes	[142,209,211]
Kaempferol	The inhibition of viral proliferation	25.57 μM of 50% inhibited concentration	No mention	Ra strain	Yes	Yes	[213]
Panax notoginseng polysaccharides	The inhibition of viral adsorption and replication	No mention	No mention	PRV XJ5 strain	Yes	No	[214]
Germacrone	The inhibition of viral proliferation	54.51 μM for Vero cells and 88.78 μM for LLC-PK-1 cells	233.5 μM for Vero cells and 184.1 μM for LLC-PK-1 cells	Variant PRV and PRV vaccine strain Barth K61	Yes	No	[206]
Plantago	The inhibition of viral attachment and penetration; decreasing ROS (reactive oxygen species) production	No mention	No mention	PRV XJ5	Yes	No	[215]
Quercetin	The inhibition of viral adsorption	2.618 ± 0.673 μM of 50% inhibited concentration	Above 599 μM	HNX strain	Yes	Yes	[216]
Isatis indigotica	The inhibition of viral proliferation	11 μg/mL	299 μg/mL	TNL strain	Yes	No	[217]
Radix isatidis	The inhibition of viral proliferation; killing virus directly	The inhibition rate of viral replication by 14.674–30.84%	No mention	Min A strain	Yes	No	[207]
Marine *Bacillus* S-12–86 lysozyme	The inhibition of viral proliferation; killing virus directly	0.46 mg/L	100 mg/L	Min A strain	Yes	No	[218]
Diammonium glycyrrhizin	Killing virus directly	No mention	Above 1250 μg/mL	Bartha K-61	Yes	No	[219]
Vanadium-substituted Heteropolytungstate	Killing virus directly	3.5–5.0 mg/L	400–420 mg/L	Bartha strain	Yes	No	[220]
Graphene Oxide	Killing virus directly	No mention	No mention	HNX strain	Yes	No	[223]
Ivermectin	The inhibition of viral DNA polymerase UL42 in entering the nucleus	No mention	No mention	No mention	Yes	Yes	[208]
Phosphonoformate sodium	Inhibition of viral DNA polymerase	Nearly 60 μg/mL of 50% inhibited concentration	No mention	Kaplan	Yes	No	[221]

### 6.3. Novel Small RNAs

Owing to the characteristic of targeting mRNA degradation, small RNAs are extensively used for searching gene functions and are also considered a novel therapeutic approach that effectively inhibits viral replication and interferes in protein synthesis, including small interfering RNAs (siRNAs) and microRNAs (miRNAs) [224,225,226]. In a previous study, it was proved that the PRV processivity factor UL42 is critical for viral replication and can improve the catalytic activity of the DNA polymerase, suggesting that it may be a latent drug target for antiviral treatment against PRV infection [2]. To verify this guess, three siRNAs (siR-386, siR-517, and siR-849) directed against UL42 were synthesized and defined their anti-PRV activities in cell culture, and the results showed that these three siRNAs induce great inhibitory effects on UL42 expression after PRV infection and impair viral replication [227]. miR-21 is one of the earliest miRNAs to be discovered, and it is associated with the immune response, viral replication, cell apoptosis and cancer [228,229,230]. In Huang’s study, it demonstrated that miR-21 plays a crucial role in the immune response to PRV infection and can directly target interferon-γ inducible protein-10 (IP-10) to inhibit PRV replication in PK15 cells [230]. Subsequently, the detailed function of a large latency transcript (LLT) miRNA cluster was further studied, and PRV-encoded prv-miR-LLT11a appeared during initial downregulation and following upregulation in PK15 cells with PRV-infection, suggesting that it may have obviously repressed viral replication [231]. However, their potential mechanism remains unclear so far, which needs deeper research.

## 7. Conclusions and Future Perspectives

It has been nearly 200 years since PR was first discovered in 1813 in America, and PRV infection has become one of the most important pathogens leading to reproductive failure of pregnant sows, nervous disorders in newborn piglets, and respiratory distress in growing pigs [2]. Although few countries have eradicated PR from their swine populations due to the application of gE-deleted PRV vaccines and the DIVA strategy, PR is still prevalent in most countries, especially in China, which has resulted in huge economic losses to the swine industry during the past several decades [157]. With continuous exploration, the roles of the various components, pathogenesis, diagnosis method and prevention of PRV are being developed in depth. In the past decades, the infected host of PRV has ranged from various animals to human beings, including pigs, dogs, cats, cattle, sheep, goats, captive mink, wild foxes, captive foxes, wolves and lynxes [232]. PRV infection invades the peripheral nervous system in pigs with the occasional invasion of the central nervous system after primary infection at mucosal surfaces [15]. For its diagnosis, different types of detection technology have been generated, such as serological approaches, molecular biology approaches, and other approaches (VI and the paper biosensor doped with Fe_3_O_4_@SiO_2_–NH_2_ and MWCNTs). As one of the serological assays, ELISA is most widely used in detecting the infection of PRV wild strains due to its sensitive and rapid features [1]. It is worth nothing that PRV infection in humans is confirmed by high-through sequencing technology, which is time-consuming and expensive [6,138]. Thus, these problems should be the focus of future efforts.

In terms of the prevention and control of PR, vaccination with the Bartha-K61 strain is the most common method, and PR is well controlled worldwide. However, PR re-emerged at the end of 2011 in Chinese pig herds which were vaccinated with the Bartha-K61 strain, and it was demonstrated that re-emerging PR is caused by variant PRV strains, suggesting that the commercial vaccine containing the Bartha-K61 strain cannot provide full protection against the variant PRV challenge [111]. With continuous research, various types of vaccine candidates have been created, such as inactivated vaccines, live gene-deleted vaccines, live attenuated recombinant vaccines, DNA vaccines and subunit vaccines, though the last two types of vaccines are rarely used in the clinical prevention of PRV. At present, PRV vaccines that are widely used in pig farms are mainly the licensed live attenuated gene-deleted vaccines, but they may appear to cause viral virulence reversion and influence the safety of vaccinated pigs. Another disadvantage is that these commercial attenuated vaccines can lead to various susceptibility and immune responses in some species (goat, dog and mink) [99]. Therefore, it is very necessary to develop safer vaccines with efficient protection against PRV infection in the future.

In conclusion, the current study and clinical progress data on PR prevention and control are optimistic, and we believe that we can achieve the goal of eradicating PR worldwide in the near future.

## Figures and Tables

**Figure 1 viruses-14-01638-f001:**
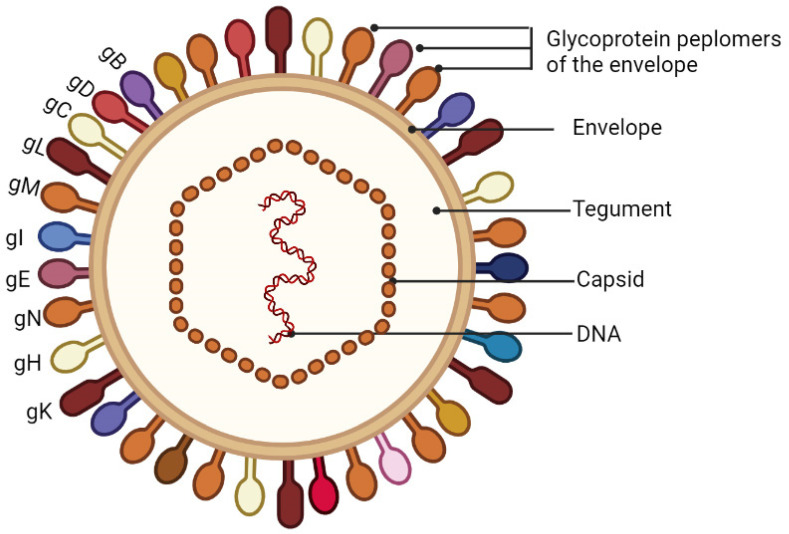
Schematic diagram of the PRV virion. PRV virions are composed of four structural elements, including a linear double-stranded DNA genome, an icosahedral protein capsid, a protein tegument layer, and a lipid envelope containing viral glycoproteins.

**Figure 2 viruses-14-01638-f002:**
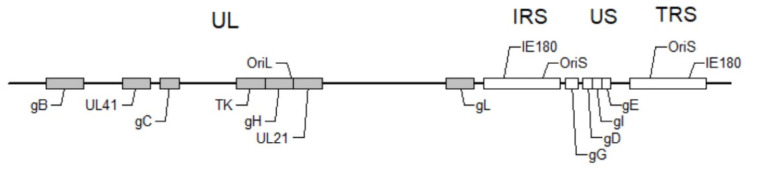
A map of the PRV genome showing the location of PRV genes discussed in this review. The PRV genome consists of the unique long region (UL) and unique short region (US), which is flanked by the internal (IRS) and terminal (TRS) repeat sequences. The genes represented by gray boxes locate in the UL, including gB, UL41, gC, TK, gH, UL21 and gL; the genes represented by white boxes locate in the US, including gG, gD, gI and gE.

**Figure 3 viruses-14-01638-f003:**
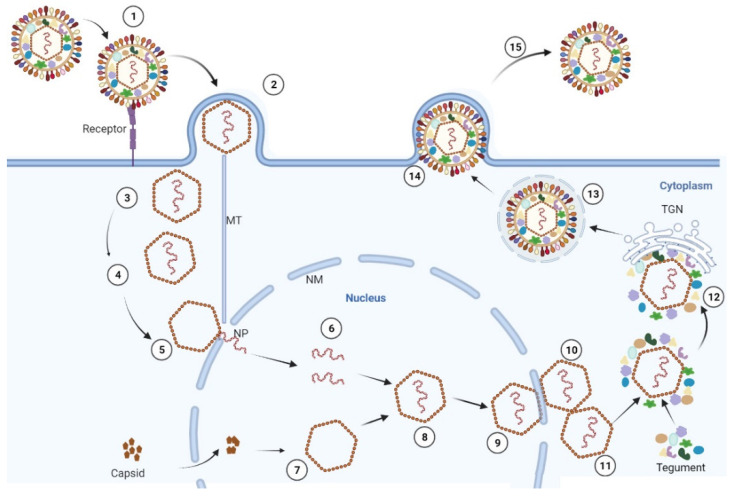
The replication cycle of PRV. After adsorption (1) and penetration (2), capsids are transported to the nucleus (3) via interaction with microtubules (MT) (4), docking at the nuclear pore (NP) (5) where the viral genome is released into the nucleus. In the nucleus, DNA replications occur (6). The capsid proteins are transported to the nucleus and are assembled around a scaffold (7), and then are assembled into a nucleocapsid with the insertion of the genomic DNA (8). The nucleocapsid leaves the nucleus by budding at the inner nuclear membrane (INM) (9), followed by fusion of the envelope of these primary virions located in the perinuclear space (10) with the outer nuclear membrane (11). Final maturation then occurs in the cytoplasm by the secondary envelopment of intra-cytosolic capsids via budding into vesicles of the trans-Golgi network (TGN) (12) containing viral glycoproteins, resulting in an enveloped virion within a cellular vesicle (13). After transport to the cell surface (14), vesicle and plasma membranes fuse, releasing a mature, enveloped virion from the cell (15).

**Figure 4 viruses-14-01638-f004:**
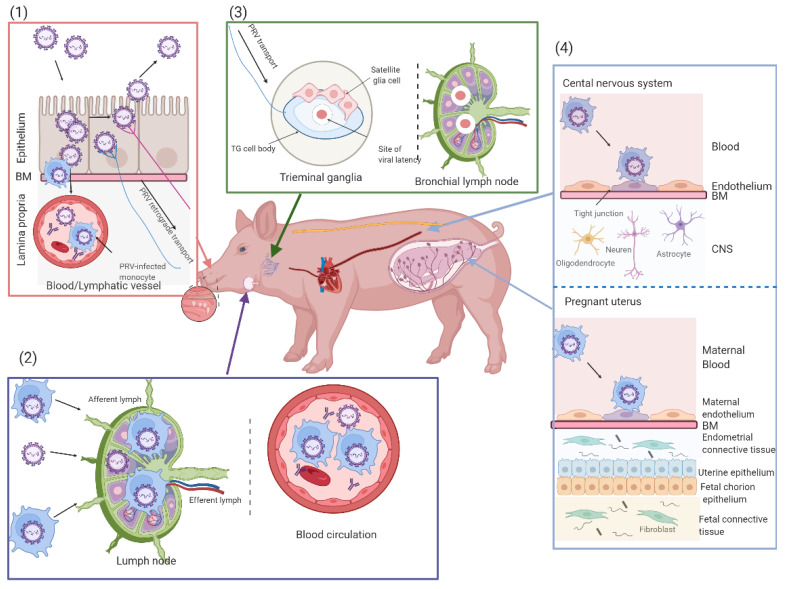
Schematic representation of the pathogenesis of PRV in pigs in different stages of growth. (1) Primary viral replication in the epithelial cells (ECs) of the upper respiratory tract: PRV first infects epithelial cells, with a viral spread and shedding, and then crosses the basement membrane (BM) and lamina propria by using single infected leukocytes to reach the blood circulation and draining lymph nodes. Lastly, PRV entry occurs at nerve endings of the peripheral nervous system and diffuses retrogradely to trigeminal ganglia (TG). (2) PRV replication in the draining lymph nodes and cell-associated viremia. (3) Establishment of PRV latency in the trigeminal ganglia (TG) neurons. (4) Secondary replication in target organs (the pregnant uterus and the central nervous system (CNS)): the secondary replication in the ECs of the pregnant uterus can lead to vasculitis and multifocal thrombosis, with an abortion of sows, and in newborn piglets, sudden death usually occurs in the absence of clinical signs.

**Figure 5 viruses-14-01638-f005:**
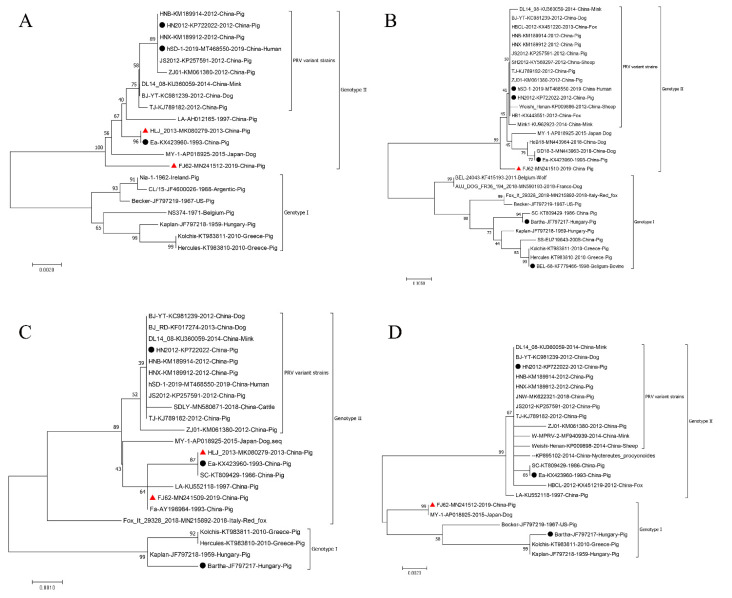
Phylogenetic trees based on the nucleotide sequences of the gE (**A**), gC (**B**), gD (**C**) and gB (**D**) genes of PRV strains from different hosts/regions using the neighbour-joining method with a bootstrap test of 1000 replicates using MEGA 7.0 software (www.megasoftware.net; Access date: 15 April 2022). The hosts, countries, years, names and GenBank accession numbers of the reference strains employed in this phylogenetic tree are labeled. Black circles and red triangles represent vaccine strains and recombinant strains, respectively.

**Figure 6 viruses-14-01638-f006:**
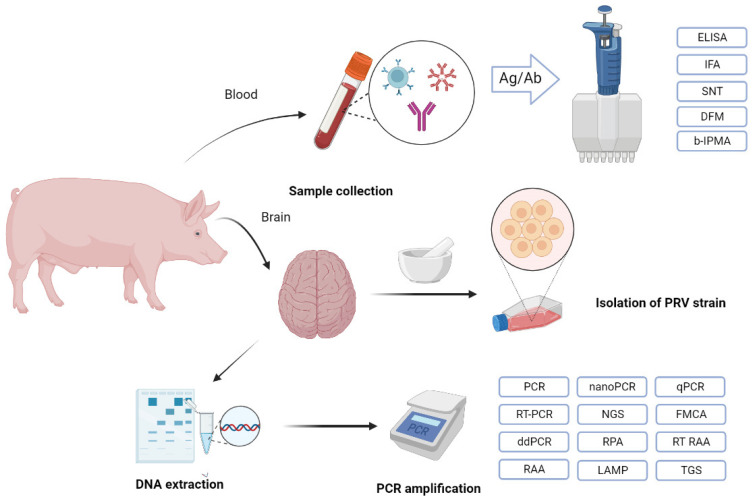
Flowchart of common diagnostic methods for PRV infection. Two types of common methods are used to verify the PRV infection based on the PRV-specific antibodies and nucleic acids. Among them, serological approaches for the detection of PRV infection include the enzyme-linked immunosorbent assays (ELISA), indirect immunofluorescence assay (IFA), serum neutralization test (SNT), direct-immunofluorescence method (DFM) and blocking immunoperoxidase monolayer assay (b-IPMA). Molecular biology approaches include polymerase chain reaction (PCR), real-time PCR (RT-PCR), TaqMan real-time PCR (qPCR), nano PCR, droplet digital PCR (ddPCR), real-time recombinase-aided amplification (RT RAA), loop-mediated isothermal amplification (LAMP), real-time fluorescent detection (real-time RPA assay), duplex fluorescence melting curve analysis (FMCA), next-generation sequencing (NGS), probe-based fluorescence melting curve analysis (FMCA), real-time recombinase-aided amplification assay (RT RAA) and third-generation sequencing (TGS).

**Table 1 viruses-14-01638-t001:** Comparison of diagnostic methods of PRV infection.

	Molecular	Serology
Test type	Viral	Antibody
Description	Nucleic acid amplification test to detect viral DNA	Detects the presence of IgA, IgM/IgG antibodies against PRV
Platform technology	PCR, RT-PCR, LAMP, qPCR, ddPCR, FMCA	ELISA, SNT, IFA, IPMA, DFM
Sample type	Brains, Hearts, livers, spleens, lungs, kidneys and lymph nodes	Plasma, serum, whole blood
Result turnaround time	<5 h	15–30 min

**Table 2 viruses-14-01638-t002:** List of molecular diagnostic methods of PRV infection.

Name of Diagnostic Assay	Sensitivity	Target Gene	Turnaround Time	Samples Used	References
Conventional polymerase chain reaction (PCR)	——	gE gene	Result in <5 h	Various tissue	[119,121]
Duplex droplet digital PCR (ddPCR) assary	4.75 copies/µL	Both gE and gB genes	Result in <2 h	Lung, brain, liver and spleen	[124]
SYBR green I-based duplex real-time PCR assay	37.8 copies/μL	gE gene	Result within 50 min	Hearts, livers, spleens, lungs, kidneys, brains and lymph nodes	[118]
Real-time recombinase-aided amplification assay (RAA)	Three 50% TCID_50_	gE gene	Result in 75 min	Lung, lymph node, tonsil and spleen	[125]
Triplex real-time PCR	0.5 TCID_50_ for classical strains, 0.2 TCID_50_ for variant strains and 0.05 TCID_50_ for vaccine strains	gE and gI genes	Result within 1 h	PRV strains	[127]
Probe-based fluorescence melting curve analysis (FMCA)	1 × 10^0^ copies per reaction	gC and gE genes	Result in <2 h	PRV strains	[120]
Loop-mediated isothermal amplification (LAMP) assay	10 copies per sample	gE and gG genes	Result in <2 h	PRV strains and clinical tissue samples	[122]
Duplex nanoparticle-assisted polymerase chain reaction (nanoPCR)	6 copies/μL	gE gene	Result in 80 min	The recombinant plasmids pET30a-PRV-gE and pUC57-PBoVNS1	[128]
Real-time quantitative PCR (RT-qPCR)	Oral fluid of 53% and nasal swab of 70%	gB gene	Result in <1 h	Oral fluid and nasal swab	[129]
Metagenomic next-and third-generation sequencing (mNGS/TGS)	——	Short- and long-read sequencing	——	Brains	[137]
Real-time fluorescent detection (real-time RPA assay)	100 copies per reaction	gD gene	Result within 20 min	Tissue	[130]
Lateral flow dipstick (RPA LFD assay)	160 copies per reaction	gD gene	Result within 20 min	Tissue	[130]
Magnetic beads-based chemiluminescent assay	100 μmol/5 pM	——	Result in 20 min	Serum samples	[131]

**Table 3 viruses-14-01638-t003:** Overview of genetically modified live PRV strains for vaccination in pigs.

Gene-Deleted Vaccines	Vaccine Strains	Progenitor Strains	Deleted Gene	Technology Used	Authorization	References
Single gene-deleted vaccine	Omnivac	BUK	TK gene	Natural losses	Licensed	[27]
	2.4N3A	NIA-3 (field strain)	gE gene	HR	Licensed	[166]
	PRV(LA-A^B^)	AH02LA (field strain)	gE gene	BCA	Not available	[167]
	HN1201ΔgE (inactivated)	HN1201 (field strain)	gE gene	HR	Licensed	[154]
	rPRVTJ-delgE	TJ (field strain)	gE gene	HR	Not available	[11]
Double gene-deleted vaccine	Omnimark	Omnivac (BUK)	TK and gIII genes	Natural losses	Licensed	[145]
	Begonia	2.4N3A (NIA-3)	TK and gE genes	Natural losses	Licensed	[168]
	NIA3-783	2.4N3A (NIA-3)	TK and gE genes	HR	Licensed	[146]
	Tolvi	field strain	TK and gpX genes	HR	Licensed	[169]
	D1200/D560	NIA-3	gD and gI genes	HR	Not available	[148]
	AD-YS400	Yangsan (field strain)	TK and gE genes	HR	Not available	[170]
	JS-2012-ΔgE/gI	JS-2012 (field strain)	gE and gI genes	HR	Not available	[171]
	gE-TK-PRV	TNL (field strain)	TK and gE genes	HR	Not available	[172]
	vZJ01ΔgE/gI (inactivated)	ZJ01 (field strain)	gE and gI genes	BCA	Not available	[157]
	PRV (PRV^ΔTK&gE-AH02^)	AH02LA (field strain)	TK and gE genes	HR	Not available	[173]
Triple gene-deleted vaccine	6C2	Field strain	TK, gE and gI genes	HR	Not available	[174]
	SA215	Fa (classical strain)	gE, gI and TK genes	HR	Licensed	[175]
	rSMXΔgI/gEΔTK	Field strain	TK, gE and gI genes	HR	Not available	[176]
	rPRVTJ-delgE/gI/TK-	rPRVTJ-delgE (TJ strain)	TK, gE and gI genes	HR	Not available	[29]
	vPRV HN1201	HN1201 (field strain)	TK, gE and gI genes	HR	Not available	[158]
	gE^-^/gI^-^/TK^-^ PRV	HeN1 (field strain)	TK, gE and gI genes	CRISPR/Cas9	Not available	[177]
	rPRV NY-gE^−^/gI^−^/TK^−^	NY (field strain)	TK, gE and gI genes	HR and CRISPR/Cas9	Not available	[155]
		201715 (field strain)	gE, gC and TK genes	CRISPR/Cas9	Not available	[178]
	rPRV/XJ5-gE^−^/gI^−^/TK^−^	XJ5 (field strain)	gE, gI and TK genes	HR	Not available	[159]
	rGXΔTK/gE/gI	GX (field strain)	TK, gE and gI genes		Not available	[165]
Four gene-deleted vaccine	PrV (376)	PrV (376)	gD, gG, gI and gE genes		Not available	[147]
	——	C (field strain)	gI, gE, Us9 and Us2 genes	Natura losses	Licensed	[179]
	PRV GDFS-delgI/gE/US9/US2	GDFS (field strain)	gI, gE, Us9 and Us2 genes	CRISPR/Cas9	Not available	[161]
	rZJ01-ΔgI/gE/TK/UL13	ZJ01	gI, gE, TK and UL13 genes	CRISPR/Cas9	Not available	[29]
Five gene-deleted vaccine	PRV rHN1201^TK−/*gE*−/*gI*−/11*k*−/28*k*−^	HN1201 (field strain))	TK, gI, gE, 11k and 28k genes	BCA	Not available	[164]

Note: HR is the homologous DNA recombination; BCA is the bacterial artificial chromosome; CRISPR/Cas9 is the clustered regularly interspaced short palindromic repeats/Cas9.

## Data Availability

Not applicable.

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
