# Peer review of "Pseudorabies Virus: From Pathogenesis to Prevention Strategies"

_viruses, 2022, doi:10.3390/v14081638_

Round 1
Reviewer 1 Report
In this study, the authors gave a review on the latest clinical progress in the prevention and control of PRV infection via the development of vaccines, traditional herbal medicines and novel small RNAs. Form this review, the researchers could get a better understanding of PRV. For publication, minor revisions are requested.
1. Figure 1 Schematic diagram of the RSV virion, different viral glycoproteins should use different diagram.
2.Figure 4 did not fully present the diagnostic methods according to Table 1 and 2.
3. Table showed provided to list the Chinese herbal medicines with anti-PRV activity, including sources, inhibition rate and mechanism.
Author Response
Reviewer 1
In this study, the authors gave a review on the latest clinical progress in the prevention and control of PRV infection via the development of vaccines, traditional herbal medicines and novel small RNAs. Form this review, the researchers could get a better understanding of PRV. For publication, minor revisions are requested.
Response: We thank the reviewer for the positive comments on the scientific values of our work.
Comment 1: Figure 1 Schematic diagram of the RSV virion, different viral glycoproteins should use different diagram.
Response: Thanks for your question, and we are sincerely apologized for describing errors. Figure 1 shows actually the schematic diagram of the PRV virion, but we mistakenly wrote RSV. We have changed "RSV" to "PRV" in Figure 1 according to your suggestion.
Comment 2: Figure 4 did not fully present the diagnostic methods according to Table 1 and 2.
Response: Thank you. We are apologized for we did not fully present the diagnostic methods in Figure 4. We have revised the Figure 4 according to your suggestion.
Comment 3: Table showed provided to list the Chinese herbal medicines with anti-PRV activity, including sources, inhibition rate and mechanism.
Response: Yes, your suggestion is right and thank you very much for your advice. We have added the Table 4 to revised manuscript according to your suggestion. Table 4 showed the source, mechanism, 50% effective concentration, 50% cytotoxic concentration, PRV strain, and the experiment in vitro and in vivo.

Reviewer 2 Report
Dr. Wang focuses on the animal virus PRV and gives the information about it. In fact, several PRV related reviews have been reported, such as Lisa E Pomeranz et al., Microbiol Mol Biol Rev . 2005 and Zongyi Bo viruses, 2022. In the present review, the authors said that they firstly introduce the structural composition and life cycle of PRV virions, while these viral proteins have been described in the previous reviews. However, several key papers have been ignored, such as T Kramer et al., J Virol . 2011, Tal Kramer et al., Cell Host Microbe . 2012.
Importantly, PRV as a traditional animal virus can trans-species to infect human and induce human disease. These cases gave a potential signal that the PRV might be a emerging virus in humans. Therefore, the mechanism of virus infecting human needs to be done in the review. However, the review is lack of the related information. The authors should add more information about the infection mechanism in human, even that is a hypothesis. In addition, the virus has two IE180 copies, why?What is different between these two IE180?
Several minor points should be carefully checked in the whole manuscript, here just give two examples
Line152, the phase “in contrast” is not appropriate used in here.
Line162, the phase “IE180 3’URT”, the URT is not right , change to “UTR”
Author Response
Reviewer 2
Comment 1: Dr. Wang focuses on the animal virus PRV and gives the information about it. In fact, several PRV related reviews have been reported, such as Lisa E Pomeranz et al., Microbiol Mol Biol Rev. 2005 and Zongyi Bo viruses, 2022. In the present review, the authors said that they firstly introduce the structural composition and life cycle of PRV virions, while these viral proteins have been described in the previous reviews. However, several key papers have been ignored, such as T Kramer et al., J Virol. 2011, Tal Kramer et al., Cell Host Microbe . 2012. PRV as a traditional animal virus can trans-species to infect human and induce human disease. These cases gave a potential signal that the PRV might be a emerging virus in humans. Therefore, the mechanism of virus infecting human needs to be done in the review. However, the review is lack of the related information. The authors should add more information about the infection mechanism in human, even that is a hypothesis.
Response: Firstly, we thank the reviewer for the comments on the scientific values of our work. Secondly, we are apologized for describing error, because this paper did not the first introduction of the structural composition and life cycle of PRV virions, and we have revised this sentence in line 26 in revised manuscript. In the present overview, we have summarized the virion structure of PRV to prevention of PR, including the virion structure, the genome structure and life cycle of PRV, occurrence and development of PRV infection, viral genetic evolution, diagnostic methods of PR, and the prevention of PR. The previous studies researched the proteomic characterization of PRV extracellular virions, and the infection and transmission of PRV [1,2], and these papers are key to summary the above information of PRV, so we have added to the revised manuscript according to your suggestion. Lastly, for some cases of human infected with PRV, it suggests that the PRV may be an emerging virus in humans. Therefore, we have added the mechanism of virus infecting human in revised manuscript according to your suggestion.
References:
[1] Kramer T, Greco TM, Enquist LW, Cristea IM. Proteomic characterization of pseudorabies virus extracellular virions. J Virol. 2011 Jul;85(13):6427-41. doi: 10.1128/JVI.02253-10. Epub 2011 Apr 27. PMID: 21525350; PMCID: PMC3126529.
[2] Kramer T, Greco TM, Taylor MP, Ambrosini AE, Cristea IM, Enquist LW. Kinesin-3 mediates axonal sorting and directional transport of alphaherpesvirus particles in neurons. Cell Host Microbe. 2012 Dec 13;12(6):806-14. doi: 10.1016/j.chom.2012.10.013. PMID: 23245325; PMCID: PMC3527838.
Comment 2: The virus has two IE180 copies, why?What is different between these two IE180?
Response: Thanks for your question, and we are apologized for describing error. PRV encodes only one genuine immediate-early gene, IE180 (ICP4 homolog), and IE180 gene encodes 1460 amino acids, and has a protein of ~153 kDa in size. In addition, the virus has two IE180 copies in genome, which locate in different regions, with in the internal repeat sequences (IRS) and the terminal repeat sequences (TRS), respectively. For clearer expression, we have changed “IE180 gene locates in the IRS and TRS, encoding 1460 amino acids, with a protein of ~153 ku in size” to “IE180 gene encodes 1460 amino acids, with a protein of ~153 kDa in molecular mass” in line 153 in the revised manuscript.
Comment 3: Line152, the phase “in contrast” is not appropriate used in here.
Response: Thank you. We have deleted the phase “in contrast” in line 152 according to your suggestion.
Comment 4: Line162, the phase “IE180 3’URT”, the URT is not right , change to “UTR”.
Response: Thank you. We have changed the phase “IE180 3’URT” to “IE180 3’UTR” in line 162 according to your suggestion.

Reviewer 3 Report
Review Pseudorabies virus – Zheng et al – Viruses
This complete review presents various aspects of Pseudorabies viruses: structure and replication cycle, its development inside the host and associated symptoms, evolution of different strains, diagnostic methods, vaccines and other preventive strategies.
Even if many various aspects are presented, there is a significant weakness in the first part dedicated to structure and molecular biology of PRVs. Notably, the authors mention different viral proteins with their role in viral infection or viral replication, some in the 2.2 section (supposed to correspond to Genome and gene content) and some in the 2.3 section (supposed to correspond to the description of the PRV life cycle). However, an integrative view is missing here, and it is often not clear why a protein is mentioned in the 2.2 section (for example most of the glycoproteins) and others in the 2.3 section. A genomic map and a general picture illustrating the main steps of PRV replication cycle are required to follow more easily the thread of the manuscript. See comments below for more details.
In addition, the English is not very good. As English is not my mother tongue, I thoroughly recommend the authors to have their manuscript proofread for the English. In many cases, sentences are too long and it is difficult to follow.
l15: which is caused
l16: I think the first PR case was described/was observed 200 years ago, but how the authors can be sure it “occurred” for the first time only 200 years ago?
l18: PRV infection. It is characterized… too long sentence
l19: resulting in severe economic losses for the pig industry…
l23: in the past several years => in the last few years / in recent years
l36: please split into two sentences, too long
l42-43: idem please split into two sentences
l46: authors did not introduce what is gE before, please correct
l73: the pathogenesis of PRV infection and its molecular characteristics. Subsequently… (add the dot!)
l78-79: titles are not accurate as the virion structure is not included in the genome structure
l80-81: repetition of l. 56-57
l80: which belongs
l81: Similarly
l84: Are there data available about the complete composition of the tegument layer proteins? If yes, please provide these data (for example in a Table). Are these tegument proteins all encoded by PRV or are there also some cellular proteins?
L85: Please remove “Among them”
L86: Please provide the genome size of PRV (average), and how many proteins it encodes.
l86: a collection (space)
l87: on interacts with envelop proteins and the other is closely…
l89: do we have an idea of how many proteins are found in the virion? Are they all encoded by the virus?
l90: I cannot understand to which elements “both” refers to?? Moreover, a “structure” cannot “encode” proteins!!??! This sentence is not comprehensible. “its molecular biology”: its refers to??
Figure 1: Please correct enveloe => envelope. Please explain “RSV”.
Please indicate the 2 different layers in the tegument
This is perhaps not due to the authors, but the Figure 1 is much too large.
L96: Please provide an example of PRV genome map! Authors talk about Us, UL regions that we cannot locate. This map should also illustrate the arrangement of genes presented l97-98 and the transcript organization (l105). It could also be nice to indicate which genes are most conserved.
L103: please explain what is known on the role and nature of the IRS. Do the IRS and TRS correspond to the same repeated sequence?
L108-109: PRV has three origins of replication, with one (OriL) located in the UL region, and the others (OriS) located in the inverted repeats.
l112: sentence is too long. Proposal: “…various enzymes. Half of them are…”
l112: not clear if “them” refers to the 70-100 proteins or to “various enzymes”?
l113: the PRV genes
l114: based on their different functions: structural genes, virulence genes…
l115: can also be divided into immediate-early genes…
l117: please indicate what is TK?
l118: please indicate what is the UL23 region.
l118: region, and plays (not which)
l118: in the virulence of the virus.
l118: It was primarily…
l120: involved in re-activating the virus during the latent…
l122: in nerve cells, without affecting its immunogenicity.
l123: gE located in…
l123: please explain what is the US8 region?
l124: gE protein (instead of “Its protein”, a protein do not belong to the gene it encodes) / or “The corresponding protein…”
Page 4, from l.124 and below, this long section is a description of the different glycoproteins of the envelope and their different roles in the viral cycle or infectivity of immunity, rather that what should be expected with the 2.2 title “Genome and gene content”. The only element the authors provide for each glycoprotein encoding gene is whether it is essential or not for viral replication. Please adapt the title, or clarify with the creation of a new separated section for the description of glycoproteins.
L127: a reference is missing.
l127: invade and spread processes…
l128: exist in complex, which is often… (not needed to indicate that the complex is not covalent, as these are the most common protein-protein complexes in biology)
l129: in the cell membrane of infected cells and in the virus envelope/capsule. If capsule, please explain what is the capsule in the text.
l130: authors specify that gI is a membrane protein. One could expect that most of the glycoprotein are membrane proteins. However this was not indicated for gE protein. Please indicate this information for each glycoprotein.
L133-143: there is a mix of structural information, viral-cell info and immunological info. It is difficult to follow as there is not always a logical link within the same sentence or between two sentences. This part should be easily improved.
L144: what is a “fusion ring”? how the gB protein can be trimeric and also form a dimeric fusion ring? Is the dimeric fusion ring a complex of two trimers? Not clear…
L147: “Yet, it cannot appreciate…” not clear what does it means?
L149: please give signification of HSV (herpes simplex virus?)
L151: Where is the US4 region? Again, a genomic map is necessary. This information is useless without a map.
L152: why the authors provide the length of the gG gene and the length of the gG protein, and they do not provide theses details for the other glycoproteins? Please delete, or explain why it is important to provide these data for gG. Alternatively, a table with the major information for each glycoprotein could be useful (length, membrane protein or not, folding/oligomerization, essential for viral replication or not, immunogenic or not, involved in infectivity or not…)
L152-153: gG protein cannot belong to a secretory protein. It could belong to a larger complex. Please complete.
L158 and elsewhere: please replace ku by kDa
L159-160: IE180 is a protein. As such, it cannot be “highly similar to some regions of herpes simplex virus”! As I understand, IE180 is a transcription factor. If I am correct, please indicate this explicitly.
l161: “both the two proteins”: not clear which are these two proteins? IE180 and? Not clear the complementarity of function: which functions? Please clarify.
L163: G-quadruplex bind to small molecule TmPyp4 that can stabilize…
Please also explain what is TmPyp4 molecule and provide a reference.
L.165: please replace “PRV gene transcription” by “PRV replication cycle” or by “PRV infection” to avoid repetition of “transcription”
l.168: located in the UL region
l169 please delete “in molecular mass” (obvious)
l169-170: In my comprehension, the transactivation of a gene is similar to “promote the expression of”. This is the job for a transcription factor. Thus, this is a repetition.
I guess that EP stands for Early Protein and that IE stands for Immediate Early? Please specify this.
l.174 and 176: please homogenise writing of UL21
l175: located in the UL region, and the protein it encodes belongs to the tegument…
l176: but can be restored with pUL21 compensatory cells. Please add reference for pUL21 and/or explain how this cell line was constructed.
L181: ribonuclease activity both in vivo and in vitro, and can degrade…
l182: The vhs protein can also…
l183: please mention signification of “IRES”
l181-184: IRES is on RNA whereas eIF4H and eIF4B are proteins => not clear if vhs is a protease (thus cleaving proteins) and/or RNase (thus cleaving the IRES).
L186-196: a schematic view illustrating adsoption, penetration and entry steps of PRV infection would be very nice
L193: Then, the capsid interact (no -s)
L199-200: not clear what is the “enzyme function” mentioned here?
L200: “The early EP0 transcript is tested in 2h post infection” => not clear what the authors mean by “tested” here. Wouldn’t it be rather “is detected at 2 hpi”?
L202: transcription activators (with -s).
L209: I guess it should be “deoxyribonucleotides” instead of “ribonucleotides”?
Please add a reference for this reparation mechanism.
L209-211: please provide more information about these replication mechanisms. Which viral/host proteins are involved in the initiation and regulation of theta replication, of rolling-circle replication, and in the switch from theta to rolling-circle replication? As the DNA is linear in the virions, how the DNA is supposed to be circularized?
L215: PRV genomes can be involved
L217: two subunits of the ribonucleotide reductase (UL39/UL40). The viral genome also encodes an uracil… (split too long sentence)
l218: which both serve in viral DNA repair…
L221: please explain what is known about how capsid enters the nucleus? Does it harbour a NLS? If not known, please specify it.
l222: 69 => ??
L224: please rephrase, the nucleocapsid cannot be assembled (it is already assembled: DNA and capsid proteins together). Prefer virions => virions can be assembled and released (please use -ed!)
L230: what does “venerally” means?
l230: In this work/paper. This is not a study but a review article.
L237: PRV entry occurs at nerve endings…
l237 and l238 seem to me to be redundant
L247 and elsewhere: please mention “in vivo”, “ex vivo” and “in vitro” in italics characters
l248: I think authors could have used hpi abbreviation previously in the manuscript
l253: which normally disappear quickly.
L259: Is this trypsin-like serine protease encoded by the virus? Please specify if encoded by virus or host.
L271: which contain
l271: I do not understand to which entity “those” refers to? Please specify.
l273: can be transported
L281-284: too long sentence
l286: have a similar way of spreading to invade PNS neurons.
L304: fetus instead of fetuses
l306: generally display/harbour (instead of are observed)
L316: why the authors claim that infectious virus “could be tested”? was it not found actually?
L322: Asia instead of Asian
l335: in one subgroup => please specify which one.
l335: It turns out that PRV genotype II strains… (instead of It finds, not English)
L339: please explain what are non-natural animals?
l339-340: I do not understand how the authors can claim this conclusion according to the data presented. Please explain the rationale.
L340: with those of pigs.
Figure 3: I suppose that each virion encode one of the 4 proteins analysed in these phylogenies. So why the number of sequences vary from one panel to the other? Perhaps some viruses do not encode all 4 proteins? If this is the case, please specify this in the text.
L349: please explain the meaning of these percentages? Similarity sequence? Divergence? Other?
L352-353: aa cannot be inserted, deleted or mutated in a PRV gene!! Only aa in proteins, or nt in a gene. Thank you.
L354: at sites
l359: please delete “are found that it” (no need)
l362: and their gE, gC and gD genes were assigned to genotype II, whereas gB genes belongs to genotype I.
L365: from piglet in Sichuan of China is identical (100%) to MY-1 strain (No) from a wild boar in Japan.
L367: please correct the sentence as you cannot have higher or lower homology. Homology means that two sequences have a common ancestor. You cannot be more or less related with your sister or your mother. You are related to them. Please use “sequence identity” or “sequence similarity” here.
L369: may appear from a recombinant event…
l370: In another report, PRV HLJ-2013 was isolated from…
l371: belonging to the genotype II
l372: three viruses
l378: was isolated from a sick piglet in Hunan of China, between…
l380: I cannot understand “further verified the…”?
Figure 4: Isolation of PRV strain (remove The)
L393: split into two sentences: (b-IPMA). Molecular biology approaches include polymerase…
l393: quantitative TapMan real-time PCR (qPCR)
L401: widely used in serological approaches, and many PRV…
l403: two sentences: (DIVA). Besides, gB antibodies…
l405: can be used for the detection…
l415: targeting the gB or gE…
l416: technology based on the blocking fluorescent…
l417: less time for PRV detection
l418: vaccinated pigs whereas a commercial gE-ELISA kit is not.
l422: (99.26%) compared to a commercial …
l423: time and cost expenses.
l426: expected to become new clinical laboratory diagnostic methods for…
L430: PRV genes including gE, gI…
l432: quantitative TaqMan real-time PCR (qPCR)
l435: most frequently used approaches…
l444: not suitable for wide range clinical detection…
L451: to prevent PR. The methods mainly include virus isolation…
l459: and are time-consuming…
l460: rapid detection of PRV has been developed.
L469: hundred years. Vaccination…
l470-474: too long sentence
section 6.1: please explain on which scientific basis some vaccines are based on 1 or 2 or 3 or 4 genes?
L512: problem of police size
L527 please rephrase
l539: what DIVA refers to?
l544: clinical signs, and in adult red fox…
L557: replace “traditional Chinese medicine” by “this medicine”
l562: especially
l563: based on these bioactivities, it appears that Res…
l571-572: PRV infection, and they need to be further studied for promoting it to be an effective choose for animals…
l578: cannot be verified for the inhibition…
l578: why they cannot be verified in vivo??
L581: it was found that quercetin can indeed reduce the extent… (nothing obvious here)

Author Response
Reviewer 3
This complete review presents various aspects of Pseudorabies viruses: structure and replication cycle, its development inside the host and associated symptoms, evolution of different strains, diagnostic methods, vaccines and other preventive strategies.
Even if many various aspects are presented, there is a significant weakness in the first part dedicated to structure and molecular biology of PRVs. Notably, the authors mention different viral proteins with their role in viral infection or viral replication, some in the 2.2 section (supposed to correspond to Genome and gene content) and some in the 2.3 section (supposed to correspond to the description of the PRV life cycle). However, an integrative view is missing here, and it is often not clear why a protein is mentioned in the 2.2 section (for example most of the glycoproteins) and others in the 2.3 section. A genomic map and a general picture illustrating the main steps of PRV replication cycle are required to follow more easily the thread of the manuscript. See comments below for more details
Response: We thank the reviewer for the positive comments on the scientific values of our work. A genomic map and a general picture illustrating the main steps of PRV replication cycle have added to the revised manuscript according to your suggestion.
Comment 1: l15: which is caused.
Response: Thank you. We have changed“which caused” to “which is caused” in line 15 according to your suggestion.
Comment 2: l16: I think the first PR case was described/was observed 200 years ago, but how the authors can be sure it “occurred” for the first time only 200 years ago?
Response: Thank you, and we are apologized for describing errors. The first PR case was actually described 190 years ago, because it was first descried in America as early as 1813. We have changed“more than 200 years” to “been nearly 200 years” in line 15 according to your suggestion.
Comment 3: l18: PRV infection. It is characterized… too long sentence.
Response: Thank you. We have changed“PRV infection, which is characterized” to “PRV infection. It is characterized” in line 18 according to your suggestion.
Comment 4: l19: resulting in severe economic losses for the pig industry…
Response: Thank you. We have changed“PRV infection, which is characterized” to “PRV infection. It is characterized” in line 18 according to your suggestion.
Comment 5: l23: in the past several years => in the last few years / in recent years.
Response: Thank you. We have changed“in the past several years” to “in recent years” in line 23 according to your suggestion.
Comment 6: l36: please split into two sentences, too long
Response: Thank you. We have split into two sentences according to your suggestion. It has revised to “Its etiological agent is pseudorabies virus (PRV), which has a wide range of hosts, among them, pigs are the natural host and reservoir of virus. It displays different symptoms at distinct growth phases after being infected with PRV, including the re-productive failure of sows, fatal encephalitis and 100% mortality of newborn pigs, res-piratory distress and growth block of young pigs” in line 36 according to your suggestion.
Comment 7: l42-43: idem please split into two sentences
Response: Thank you. We have split into two sentences according to your suggestion. It has revised to “In addition, PRV infection might cause the endophthalmitis and encephalitis of human beings. Some studies determined the presence of PRV specific sequences in the pa-tients’ tissues using metagenomic next-generation sequencing, and a human-originated PRV strain hSD-1/2019 was isolated from the cerebrospinal fluid of patient with acute encephalitis”.
Comment 8: l46: authors did not introduce what is gE before, please correct
Response: Thank you. We have added the introduction of gE to the revised manuscript in line 46 according to your suggestion.
Comment 9: l73: the pathogenesis of PRV infection and its molecular characteristics. Subsequently… (add the dot!)
Response: Thank you. We have changed“the pathogenesis of PRV infection and the molecular characteristic. Subsequently” to “the pathogenesis of PRV infection and its molecular characteristics. Subsequently” in line 73 according to your suggestion.
Comment 10: l78-79: titles are not accurate as the virion structure is not included in the genome structure.
Response: Thank you. We have revised the title in line 48 in the revised manuscript according to your suggestion.
Comment 11: l80-81: repetition of l. 56-57.
Response: Thank you. We have deleted this sentence in line 80-81 in revised manuscript according to your suggestion.
Comment 12: l80: which belongs.
Response: Thank you. We have deleted this sentence in line 80 in revised manuscript according to your suggestion.
Comment 13: l81: Similarly.
Response: Thank you. We have changed “Similar” to “Similarly” in line 81 in revised manuscript according to your suggestion.
Comment 14: l84: Are there data available about the complete composition of the tegument layer proteins? If yes, please provide these data (for example in a Table). Are these tegument proteins all encoded by PRV or are there also some cellular proteins?
Response: Yes, your suggestion is right, and thank you very much for your advice. But the reasons are those why we did not put Table into this overview as follows: (1) in this overview, we aimed to describe some genes which is related to the viral virulence, such as gE, gI, TK, because PRV strain can be vaccine candidates after deleting these genes. For other genes, we can briefly summary in this overview. (2) This overview has added Table and Figure, causing a lot of Table and Figure in revised manuscript. (3) The tegument layer proteins of PRV virion were summarized in previous study. Therefore, we did not put Table (the summary of tegument layer proteins of PRV virion) into the revised manuscript. For the convenience of viewing, we summarized these tegument layer proteins of PRV virion in the following Table:
|
Gene |
Size (KDa) |
Common |
Proposed functions |
|
UL51 |
25 |
- |
Viral egress (secondary envelopment); tegument protein, potentially palmytoilated |
|
UL49 |
25.9 |
VP22 |
Interacts with C-terminal domains of gE & gM; tegument protein |
|
UL48 |
45.1 |
VP16, α-TIF |
Gene regulation (transactivator); viral egress (secondary envelopment); tegument protein |
|
UL47 |
80.4 |
VP13/14 |
Viral egress (secondary envelopment); tegument protein |
|
UL46 |
75.5 |
VP11/12 |
Unknown; tegument protein |
|
UL31 |
30.4 |
- |
Viral egress (nuclear egress); present only in primary enveloped virion; interacts with UL34 |
|
UL36 |
324.4 |
VP1/2 |
Viral egress (capsid tegumentation); large tegument protein; interacts with UL37 and capsid |
|
UL37 |
98.2 |
- |
Viral egress (capsid tegumentation); interacts with UL36 |
|
UL41 |
40.1 |
VHS |
Gene regulation, RNAse, degrades host and viral mRNAs |
|
UL21 |
16.7 |
- |
Unknown, capsid-associated tegument protein; interacts with UL16 |
|
UL16 |
34.8 |
- |
Unknown; tegument protein; interacts with UL21 |
|
UL13 |
41.1 |
VP18.8 |
Unknown; protein-serine/threonine kinase |
|
UL11 |
7 |
- |
Viral egress (secondary envelopment); membrane-associated tegument protein |
|
US3 |
36.9 |
PK |
Viral egress (nuclear egress); inhibits apoptosis; major form of protein kinase (41-kDa mobility); found in both primary and secondary enveloped virions |
|
US2 |
27.7 |
28K |
Tegument protein; membrane associated protein |
Comment 15: L85: Please remove “Among them”.
Response: Thank you. We have deleted “Among them” in line 85 in revised manuscript according to your suggestion.
Comment 16: L86: Please provide the genome size of PRV (average), and how many proteins it encodes.
Response: Thank you. We have added the genome size of PRV (average) and the number of its proteins which are encoded by PRV in line 86 in revised manuscript according to your suggestion.
Comment 17: l86: a collection (space).
Response: Thank you. We have changed “acollection” to “a collection” in line 86 in revised manuscript according to your suggestion.
Comment 18: l87: on interacts with envelop proteins and the other is closely….
Response: Thank you. We have changed “one that” to “the other” in line 87 in revised manuscript according to your suggestion.
Comment 19: l89: do we have an idea of how many proteins are found in the virion? Are they all encoded by the virus?
Response: Thanks for your questions. There are 72 proteins which are encoded by PRV, and it has added into the line 86 in the revised manuscript according to your comment 16.
Comment 20: l90: I cannot understand to which elements “both” refers to?? Moreover, a “structure” cannot “encode” proteins!!??! This sentence is not comprehensible. “its molecular biology”: its refers to??
Response: Thanks for your question, and we are sincerely apologized for describing errors. In fact, we originally want to express that the envelope of PRV virion contains various glycoproteins and membrane proteins, and the tegument and capsid of PRV virion also collection of some proteins. To avoid confusing readers, we have deleted this sentence in line 90 in the revised manuscript.
Comment 21: Figure 1: Please correct enveloe => envelope. Please explain “RSV”.
Please indicate the 2 different layers in the tegument(此问题不懂,未修改)
This is perhaps not due to the authors, but the Figure 1 is much too large.
Response: Thanks for your question, and we are sincerely apologized for describing errors. We have corrected “enveloe” to “envelope”. In addition, the Figure 1 shows actually the schematic diagram of the PRV virion, but we mistakenly wrote RSV. We have changed "RSV" to "PRV" in Figure 1 according to your suggestion. At last, we are sorry the size of Figure 1, but its size was adjusted by the editors, not authors.
Comment 22: L96: Please provide an example of PRV genome map! Authors talk about Us, UL regions that we cannot locate. This map should also illustrate the arrangement of genes presented l97-98 and the transcript organization (l105). It could also be nice to indicate which genes are most conserved.
Response: Thanks for your question, we have added the PRV genome map (the Figure 2) into the line 96 in the revised manuscript according to your suggestion.
Comment 23: L103: please explain what is known on the role and nature of the IRS. Do the IRS and TRS correspond to the same repeated sequence?
Response: Thanks for your question, and we are apologized for we did not describe it clearly. IRS refers to internal repeat sequences, and TRS refers to terminal repeat sequences. The US region of viral DNA genome is flanked by the internal and terminal repeat sequences (IRS and TRS, respectively), and their sequences are different.
Comment 24: L108-109: PRV has three origins of replication, with one (OriL) located in the UL region, and the others (OriS) located in the inverted repeats.
Response: Thank you. We have changed “PRV has three origins of replication, containing one of OriL located in the UL region and two copies of OriS in the inverted repeats” to “PRV has three origins of replication, with one of OriL located in the UL region, and the others (OriS) located in the inverted repeats” in line 108-109 in revised manuscript according to your suggestion.
Comment 25: l112: sentence is too long. Proposal: “…various enzymes. Half of them are…”.
Response: Thank you. We have revised “various enzymes, and among them, half of them” to “various enzyme. Half of them” in line 112 in the revised manuscript according to your suggestion.
Comment 26: l112: not clear if “them” refers to the 70-100 proteins or to “various enzymes”?
Response: Thanks for your question, and we are apologized for we did not describe it clearly. “Them” refers to the 70-100 proteins in line 112 in the manuscript.
Comment 27: l113: the PRV genes.
Response: Thank you. We have changed “the PRV gene” to “the PRV genes” in line 113 in the revised manuscript according to your suggestion.
Comment 28: l114: based on their different functions: structural genes, virulence genes….
Response: Thank you. We have changed “based on its different functions, such as structural genes” to “based on their different functions: structural genes, virulence genes….” in line 114 in the revised manuscript according to your suggestion.
Comment 29: l115: can also be divided into immediate-early genes…
Response: Thank you. We have changed “can also be divided by immediate-early genes” to “can also be divided into immediate-early genes…” in line 115 in the revised manuscript according to your suggestion.
Comment 30: l117: please indicate what is TK?
Response: Thanks for your question, and we have indicated what is TK in line 117 in the revised manuscript according to your suggestion.
Comment 31: l118: please indicate what is the UL23 region.
Response: Thanks for your question, and we are apologized for we did not describe it clearly. PRV UL23 gene is also named as thymidine kinase (TK). This sentence has corrected in line 118 in the revised manuscript.
Comment 32: l118: region, and plays (not which)
Response: Thank you. We have corrected this sentence in line 118 in the revised manuscript according to your suggestion.
Comment 33: l118: in the virulence of the virus.
Response: Thank you. We have corrected this sentence to “PRV thymidine kinase (TK), namely UL23 gene, plays a decisive role in the virulence of the virus.” in line 118 in the revised manuscript according to your suggestion.
Comment 34: l118: It was primarily…
Response: Thank you. We have changed “It is primarily…” to “It was primarily…” in line 118 in the revised manuscript according to your suggestion.
Comment 35: l120: involved in re-activating the virus during the latent…
Response: Thank you. We have changed “involved in activating virus in the latent” to “involved in re-activating the virus during the latent…” in line 120 in the revised manuscript according to your suggestion.
Comment 36: l122: in nerve cells, without affecting its immunogenicity.
Response: Thank you. We have changed “in nerve cells, and cannot affect its immunogenicity” to “in nerve cells, without affecting its immunogenicity” in line 122 in the revised manuscript according to your suggestion.
Comment 37: l123: gE located in…
Response: Thank you. We have changed “gE gene locating in” to “gE located in…” in line 123 in the revised manuscript according to your suggestion.
Comment 38: l123: please explain what is the US8 region?
Response: Thanks for your question, and we are apologized for we did not describe it clearly. PRV gE gene is also named as US8 gene, and it locates in the US region. This sentence has corrected in line 123 in the revised manuscript.
Comment 39: l124: gE protein (instead of “Its protein”, a protein do not belong to the gene it encodes) / or “The corresponding protein…”
Response: Thank you. We have changed “Its protein” to “gE protein” in line 124 in the revised manuscript according to your suggestion.
Comment 40: Page 4, from l.124 and below, this long section is a description of the different glycoproteins of the envelope and their different roles in the viral cycle or infectivity of immunity, rather that what should be expected with the 2.2 title “Genome and gene content”. The only element the authors provide for each glycoprotein encoding gene is whether it is essential or not for viral replication. Please adapt the title, or clarify with the creation of a new separated section for the description of glycoproteins.
Response: Thanks for your question, and we are apologized for we did not describe it clearly. We have revised the title in the revised manuscript according to your suggestion.
Comment 41: L127: a reference is missing.
Response: Thank you. We have added a reference in line 127 in the revised manuscript according to your suggestion.
Comment 42: l127: invade and spread processes…
Response: Thank you. We have changed “invade and spread process” to “invade and spread processes…” in line 127 in the revised manuscript according to your suggestion.
Comment 43: l128: exist in complex, which is often… (not needed to indicate that the complex is not covalent, as these are the most common protein-protein complexes in biology)
Response: Thank you. We have changed “exist in a complex in the form of non-covalent bond, which is often” to “exist in complex, which is often…” in line 128 in the revised manuscript according to your suggestion.
Comment 44: l129: in the cell membrane of infected cells and in the virus envelope/capsule. If capsule, please explain what is the capsule in the text.
Response: Thanks for your question, and we are apologized for describing error. We have changed “in the cell membrane and virus capsule of infected cells” to “in the cell membrane of infected cells and in the virus envelope” in line 129 in the revised manuscript according to your suggestion.
Comment 45: l130: authors specify that gI is a membrane protein. One could expect that most of the glycoprotein are membrane proteins. However this was not indicated for gE protein. Please indicate this information for each glycoprotein.
Response: Thanks for your question. We have indicated that gE glycoprotein is membrane proteins in line 130 in the revised manuscript according to your suggestion. gI protein is a membrane protein, which can not only promote the secretion of gE glycoprotein in the endoplasmic reticulum and ensure its correct glycosylation, but also promote the transmission of virus between cells.
Comment 46: L133-143: there is a mix of structural information, viral-cell info and immunological info. It is difficult to follow as there is not always a logical link within the same sentence or between two sentences. This part should be easily improved.
Response: Thanks for your question, and we are apologized for not describing clear. We have improved this sentence in line 133-143 in the revised manuscript according to your suggestion.
Comment 47: L144: what is a “fusion ring”? how the gB protein can be trimeric and also form a dimeric fusion ring? Is the dimeric fusion ring a complex of two trimers? Not clear…
Response: Thanks for your question, and we are apologized for we did not describe it clearly. We have changed “fusion ring” to “fusion loops (FLs)”, and we have alos improved this sentence in line 144 in the revised manuscript according to your suggestion.
Comment 48: L147: “Yet, it cannot appreciate…” not clear what does it means?
Response: Thanks for your question, and we are apologized for we did not describe it clearly. We have changed “Yet, it cannot appreciate…” to “Yet, it cannot participate in…” in line 147 in the revised manuscript.
Comment 49: L149: please give signification of HSV (herpes simplex virus?)
Response: Thank you. We have added the signification of HSV to line 149 in the revised manuscript according to your suggestion.
Comment 50: L151: Where is the US4 region? Again, a genomic map is necessary. This information is useless without a map.
Response: Thank you, and we are apologized for describing errors. PRV US8 gene is also named as gG gene. This sentence has corrected in line 151 in the revised manuscript. In addition, a genomic map has added into the revised manuscript.
Comment 51: L152: why the authors provide the length of the gG gene and the length of the gG protein, and they do not provide theses details for the other glycoproteins? Please delete, or explain why it is important to provide these data for gG. Alternatively, a table with the major information for each glycoprotein could be useful (length, membrane protein or not, folding/oligomerization, essential for viral replication or not, immunogenic or not, involved in infectivity or not…)
Response: Thanks for your suggestion. We have deleted these details in line 152 in revised manuscript according to your suggestion.
Comment 52: L152-153: gG protein cannot belong to a secretory protein. It could belong to a larger complex. Please complete.
Response: Thank you, and we are apologized for describing errors. We have corrected this sentence in line 152-153 in the revised manuscript according to your suggestion.
Comment 53: L158 and elsewhere: please replace ku by kDa.
Response: Thank you. We have change “ku” to “kDa” in line 158 in the revised manuscript according to your suggestion.
Comment 54: L159-160: IE180 is a protein. As such, it cannot be “highly similar to some regions of herpes simplex virus”! As I understand, IE180 is a transcription factor. If I am correct, please indicate this explicitly.
Response: Thanks for your question. According to the study of literatures, the immediately early protein IE180 has a high level of similarity to the IE proteins of other alphaherpesviruses such as ICP4 of HSV-1, IE140 of varicella-zoster virus (VZV), IE1 of equine herpesvirus 1, and p180 of bovine herpesvirus 1 [1,2]. And we have added the references in line 159-160 in the revised manuscript.
[1] Lerma, L., et al., Partial complementation between the immediate early proteins ICP4 of herpes simplex virus type 1 and IE180 of pseudorabies virus. Virus Res, 2020. 279: p. 197896.
[2] C. Vlcek, V. Paces, M. Schwyzer. Nucleotide sequence of the pseudorabies virus immediate early gene, encoding a strong transactivator protein. Virus Genes, 2 (1989), pp. 335-346.
Comment 55: l161: “both the two proteins”: not clear which are these two proteins? IE180 and? Not clear the complementarity of function: which functions? Please clarify.
Response: Thanks for your question, and we are apologized for we did not describe it clearly. These two proteins refer to PRV IE180 and HSV I ICP4, partial complementation are found between the immediate early proteins ICP4 of herpes simplex virus type 1 and IE180 of pseudorabies virus.
Comment 56: L163: G-quadruplex bind to small molecule TmPyp4 that can stabilize…Please also explain what is TmPyp4 molecule and provide a reference.
Response: Thank you. We have added the signification of TmPyp4 molecule and a reference to line 163 in the revised manuscript according to your suggestion.
Comment 57: L.165: please replace “PRV gene transcription” by “PRV replication cycle” or by “PRV infection” to avoid repetition of “transcription”
Response: Thank you. We have changed “PRV gene transcription” to “PRV replication cycle” in line 165 in the revised manuscript according to your suggestion.
Comment 58: l.168: located in the UL region
Response: Thank you. We have changed “locating in UL region” to “located in the UL region” in line 168 in the revised manuscript according to your suggestion.
Comment 59: l169 please delete “in molecular mass” (obvious)
Response: Thank you. We have deleted “in molecular mass” in line 169 in the revised manuscript according to your suggestion.
Comment 60: l169-170: In my comprehension, the transactivation of a gene is similar to “promote the expression of”. This is the job for a transcription factor. Thus, this is a repetition.
I guess that EP stands for Early Protein and that IE stands for Immediate Early? Please specify this.
Response: Thanks for your suggestion. We have deleted “and promote the expression of IE180 protein” in line 169-170 in the revised manuscript according to your suggestion. EP0 protein is an early protein and IE180 is the immediate early gene, and we have improved this sentence in revised manuscript according to your suggestion.
Comment 61: l.174 and 176: please homogenise writing of UL21.
Response: Thank you. We have homogenized writing of UL21 in line 174 and 176 in revised manuscript according to your suggestion.
Comment 62: l175: located in the UL region, and the protein it encodes belongs to the tegument…
Response: Thank you. We have changed “locating in the UL region, and its protein belongs to the tegument proteins” to “The UL21 gene, located in the UL region, is a non-essential gene for PRV replication, and the protein encoded by UL21 belongs to the tegument proteins.” in line 175 in the revised manuscript according to your suggestion.
Comment 63: l176: but can be restored with pUL21 compensatory cells. Please add reference for pUL21 and/or explain how this cell line was constructed.
Response: Thank you. We have added the reference for pUL21 in line 176 in the revised manuscript according to your suggestion.
Comment 64: L181: ribonuclease activity both in vivo and in vitro, and can degrade…
Response: Thank you. We have changed “ribonuclease activity in both vivo and vitro, which can degrade” to “ribonuclease activity both in vivo and in vitro, and can degrade…” in line 181 in the revised manuscript according to your suggestion.
Comment 65: l182: The vhs protein can also…
Response: Thank you. We have changed “vhs can also” to “The vhs protein can also…” in line 182 in the revised manuscript according to your suggestion.
Comment 66: l183: please mention signification of “IRES”
Response: Thank you. We have added the signification of “IRES” in line 183 in the revised manuscript according to your suggestion.
Comment 67: l181-184: IRES is on RNA whereas eIF4H and eIF4B are proteins => not clear if vhs is a protease (thus cleaving proteins) and/or RNase (thus cleaving the IRES).
Response: Thanks for your question, and we are apologized for we did not describe it clearly. We have corrected this sentence in line 181-184 in the revised manuscript.
Comment 68: L186-196: a schematic view illustrating adsoption, penetration and entry steps of PRV infection would be very nice.
Response: Thank you. We have added a schematic in line 186-196 in the revised manuscript according to your suggestion.
Comment 69: L193: Then, the capsid interact (no -s).
Response: Thank you. We have changed “Then, the capsids interact” to “Then, the capsid interacts” in line 193 in the revised manuscript according to your suggestion.
Comment 70: L199-200: not clear what is the “enzyme function” mentioned here?.
Response: Thanks for your question, and we are apologized for we describe it error. We have deleted “enzyme function” in line 199-200 in revised manuscript according to your suggestion.
Comment 71: L200: “The early EP0 transcript is tested in 2h post infection” => not clear what the authors mean by “tested” here. Wouldn’t it be rather “is detected at 2 hpi”?
Response: Thanks for your question, and we are apologized for we did not describe it clearly. It was tested at 2 hpi, and we have changed “The early EP0 transcript is tested in 2h post infection” to “The early EP0 transcript is detected at 2 h post infection (hpi)” in line 200 in revised manuscript according to your suggestion.
Comment 72: L202: transcription activators (with -s).
Response: Thank you. We have changed “transcription activator” to “transcription activators” in line 202 in revised manuscript according to your suggestion.
Comment 73: L209: I guess it should be “deoxyribonucleotides” instead of “ribonucleotides”?
Please add a reference for this reparation mechanism.
Response: Thank you. We have changed “ribonucleotides” to “deoxyribonucleotides” in line 209 and have added a reference in revised manuscript according to your suggestion.
Comment 74: L209-211: please provide more information about these replication mechanisms. Which viral/host proteins are involved in the initiation and regulation of theta replication, of rolling-circle replication, and in the switch from theta to rolling-circle replication? As the DNA is linear in the virions, how the DNA is supposed to be circularized?
Response: Thanks for your question. We have added the information about these replication mechanisms and the schematic diagram of PRV replication cycle in line 209-211 in revised manuscript according to your suggestion.
Comment 75: L215: PRV genomes can be involved.
Response: Thank you. We have changed “PRV genomes can involve” to “PRV genomes can be involved” in line 215 in revised manuscript according to your suggestion.
Comment 76: L217: two subunits of the ribonucleotide reductase (UL39/UL40). The viral genome also encodes an uracil… (split too long sentence)
l218: which both serve in viral DNA repair…
Response: Thanks for your suggestion. We have split into two sentences in line 217, and have changed “which serve in viral DNA repair” to “which both serve in viral DNA repair…” in line 218 in revised manuscript according to your suggestion.
Comment 77: L221: please explain what is known about how capsid enters the nucleus? Does it harbour a NLS? If not known, please specify it.
Response: Thanks for your question, and we are apologized for we did not describe it clearly. We have added schematic diagram of PRV replication cycle in the revised manuscript according to your suggestion. In the replication cycle of the PRV, the capsid proteins are transported to the nucleus and then are assembled around a scaffold.
Comment 78: l222: 69 => ??
Response: Thanks for your question, and we are apologized for our mistake. We have deleted “69” in line 222 in revised manuscript.
Comment 79: L224: please rephrase, the nucleocapsid cannot be assembled (it is already assembled: DNA and capsid proteins together). Prefer virions => virions can be assembled and released (please use -ed!)
Response: Thank you. We have rephrased this sentence and changed “release” to
“released” in line 224 in revised manuscript according to your suggestion.
Comment 80: L230: what does “venerally” means?
Response: Thanks for your question, and we are apologized for we did not describe it clearly. It refers to “blood transmission”, and we have revised in line 230.
Comment 81: l230: In this work/paper. This is not a study but a review article.
Response: Thank you. We have changed “In this study” to “In this work” in line 230 in revised manuscript according to your suggestion.
Comment 82: L237: PRV entry occurs at nerve endings…
Response: Thank you. We have changed “PRV entry occurs nerve endings” to “PRV entry occurs at nerve endings…” in line 237 in revised manuscript according to your suggestion.
Comment 83: l237 and l238 seem to me to be redundant.
Response: Thanks for your question. We have revised this sentence in line 237 and 238 according to your suggestion.
Comment 84: L247 and elsewhere: please mention “in vivo”, “ex vivo” and “in vitro” in italics characters.
Response: Thank you. We have mention “in vivo”, “ex vivo” and “in vitro” in italics characters in line 247 and elsewhere in revised manuscript according to your suggestion.
Comment 85: l248: I think authors could have used hpi abbreviation previously in the manuscript.
Response: Thank you. We have changed “24 h post-inoculation” to “24 hpi” in line 248 in revised manuscript according to your suggestion.
Comment 86: l253: which normally disappear quickly.
Response: Thank you. We have changed “which normally soon disappear” to “which normally disappear quickly” in line 253 in revised manuscript according to your suggestion.
Comment 87: L259: Is this trypsin-like serine protease encoded by the virus? Please specify if encoded by virus or host.
Response: Thank you. The trypsin-like serine protease is excreted by the virus, and we have revised this sentence in line 259 in revised manuscript according to your suggestion.
Comment 88: L271: which contain.
Response: Thank you. We have changed “which contains” to “which contain” in line 271 in revised manuscript according to your suggestion.
Comment 89: l271: I do not understand to which entity “those” refers to? Please specify.
Response: Thanks for your question, and we are apologized for we did not describe it clearly. “those” refers to the nerve endings of sensory trigeminal ganglia (TG) and olfactory bulb, and other facial, parasympathetic, sympathetic nerve neurons that innervate the epithelium.
Comment 90: l273: can be transported.
Response: Thank you. We have changed “can be transport” to “can be transported” in line 273 in revised manuscript according to your suggestion.
Comment 91: L281-284: too long sentence.
Response: Thank you. We have split this sentence into two sentences in line 281-284 in revised manuscript according to your suggestion.
Comment 92: l286: have a similar way of spreading to invade PNS neurons.
Response: Thank you. We have changed “have to a similar way which invades PNS neurons” to “have a similar way of spreading to invade PNS neurons” in line 286 in revised manuscript according to your suggestion.
Comment 93: L304: fetus instead of fetuses.
Response: Thank you. We have changed “fetuses” to “fetus” in line 304 in revised manuscript according to your suggestion.
Comment 94: l306: generally display/harbour (instead of are observed).
Response: Thank you. We have changed “are observed” to “generally displays” in line 306 in revised manuscript according to your suggestion.
Comment 95: L316: why the authors claim that infectious virus “could be tested”? was it not found actually?
Response: Thanks for your question, and we are apologized for we did not describe it clearly. PRV can be detected in brain tissue samples of piglets naturally infected with PRV, and can be isolated from brain. So we have revised this sentence in line 316 according to your suggestion.
Comment 96: L322: Asia instead of Asian.
Response: Thank you. We have changed “Asian” to “Asia” in line 322 in revised manuscript according to your suggestion.
Comment 97: l335: in one subgroup => please specify which one.
Response: Thanks for your question, and we are apologized for we did not describe it clearly. “one subgroup” refer to a separate branch, in other words, Chinese PRV strains are clustered in one group, but Chinese PRV variant strains further clustered into a separate subgroup.
Comment 98: l335: It turns out that PRV genotype II strains… (instead of It finds, not English).
Response: Thank you. We have changed “It finds” to “It turns out that PRV genotype II strains…” in line 335 in revised manuscript according to your suggestion.
Comment 99: L339: please explain what are non-natural animals?
Response: Thanks for your question, and we are apologized for we did not describe it clearly. Pigs are the unique natural host and reservoir of PRV, and other animals are considered to non-natural animals of PRV infection. For easier understanding, we have revised “non-natural animals” to “other animals” in line 339.
Comment 100: l339-340: I do not understand how the authors can claim this conclusion according to the data presented. Please explain the rationale.
Response: Thanks for your question, and we are apologized for we did not describe it clearly. Because PRV strains isolated from other animals are both randomly distributed in two genotypes (I or II) (Figure 3). Combined with previous research [1], it guesses that PRV strains isolated from animals and human beings may have a similar ancestor with those of pigs. Therefore, we have added this sentence and revised the conclusion in line 339-340 in the revise manuscript.
Reference:
[1] Liu J, Chen C, Li X. Novel Chinese pseudorabies virus variants undergo extensive recombination and rapid interspecies transmission. Transbound Emerg Dis. 2020 Nov;67(6):2274-2276. doi: 10.1111/tbed.13784. Epub 2020 Aug 26. PMID: 32786133.
Comment 101: L340: with those of pigs.
Response: Thank you. We have changed “with these of pigs” to “with those of pigs” in line 340 in revised manuscript according to your suggestion.
Comment 102: Figure 3: I suppose that each virion encode one of the 4 proteins analysed in these phylogenies. So why the number of sequences vary from one panel to the other? Perhaps some viruses do not encode all 4 proteins? If this is the case, please specify this in the text.
Response: Thanks for your question, and we are apologized for we did not describe it clearly. Normally, each virion can encode 4 proteins (gE/gB/gC/gD). In general, researchers can upload complete genome (all genes) of virus to GenBank database, for example, PRV strains HNB, HN2012, HNX, JS-2012, ZJ01, TJ, Becker, Kolchis, Hercules. But some researchers only upload certain gene sequences of virus, such as PRV strains HeB18, Weishi, HB1, SH2012, HBCL-2012, BEL-24043, SS, SDLY, Bartha. Therefore, the number of sequences was difference in this review.
Comment 103: L349: please explain the meaning of these percentages? Similarity sequence? Divergence? Other?
Response: Thanks for your question, and we are apologized for we did not describe it clearly. These percentages refer to the average amino acid (aa) differences, and we have revised this sentence in line 349.
Comment 104: L352-353: aa cannot be inserted, deleted or mutated in a PRV gene!! Only aa in proteins, or nt in a gene. Thank you.
Response: Thanks for your question, and we are apologized for describing mistake. We have changed “and aa insertions, deletions” to “and nucleotide insertions, deletions” in line 352-353 in revised manuscript according to your suggestion.
Comment 105: L354: at sites.
Response: Thank you. We have changed “at sits” to “at sites” in line 354 in revised manuscript according to your suggestion.
Comment 106: l359: please delete “are found that it” (no need).
Response: Thank you. We have deleted “are found that it” in line 359 in revised manuscript according to your suggestion.
Comment 107: l362: and their gE, gC and gD genes were assigned to genotype II, whereas gB genes belongs to genotype I.
Response: Thank you. We have changed “and theses gE, gC and gD genes were separated into genotype II, with gB genes of geno-type I” to “and their gE, gC and gD genes were assigned to genotype II, whereas gB genes belongs to genotype I” in line 362 in revised manuscript according to your suggestion.
Comment 108: L365: from piglet in Sichuan of China is identical (100%) to MY-1 strain (No) from a wild boar in Japan.
Response: Thank you. We have changed “from piglet in Sichuan of China is highly homologous (100%) with MY-1 strain (No. AP018925) from a wild boar in Japan” to “from piglet in Sichuan of China is identical (100%) to MY-1 strain (No) from a wild boar in Japan” in line 365 in revised manuscript according to your suggestion.
Comment 109: L367: please correct the sentence as you cannot have higher or lower homology. Homology means that two sequences have a common ancestor. You cannot be more or less related with your sister or your mother. You are related to them. Please use “sequence identity” or “sequence similarity” here.
Response: Thanks for your question, and we are apologized for we did not describe it clearly. We have corrected this sentence, and have added the sequence identity of PRV variant strain FJ62 compared with other reference strains in line 367 in the revised manuscript according to your suggestion.
Comment 110: L369: may appear from a recombinant event…
Response: Thank you. We have changed “may appear recombinant event” to “may appear from a recombinant event…” in line 369 in revised manuscript according to your suggestion.
Comment 111: l370: In another report, PRV HLJ-2013 was isolated from…
Response: Thank you. We have changed “In a report, PRV HLJ-2013 which isolated from” to “In another report, PRV HLJ-2013 was isolated from…” in line 370 in revised manuscript according to your suggestion.
Comment 112: l371: belonging to the genotype II.
Response: Thank you. We have changed “belongs to the genotype II” to “belonging to the genotype II” in line 371 in revised manuscript according to your suggestion.
Comment 113: l372: three viruses.
Response: Thank you. We have changed “three virus” to “three viruses” in line 372 in revised manuscript according to your suggestion.
Comment 114: l378: was isolated from a sick piglet in Hunan of China, between…
Response: Thank you. We have changed “was isolated from the one sick piglet in Hunan of China between” to “was isolated from a sick piglet in Hunan of China, between…” in line 378 in revised manuscript according to your suggestion.
Comment 115: l380: I cannot understand “further verified the…”?
Response: Thanks for your question, and we are apologized for we did not describe it clearly. We have corrected this sentence to “which again confirmed the presence of recombinant event of PRV” in line 380 in revised manuscript. Because the recombinant event occurred in the genome of PRV strain in previous studies, Tan’s study again confirmed that PRV strain HN-2019 can happen to the recombinant event. In order to make it easier for everyone to understand, we have changed “further verified the presence of recombinant event of PRV” to “which again confirmed the presence of recombinant event of PRV”.
Comment 116: Figure 4: Isolation of PRV strain (remove The).
Response: Thank you. We have changed “The isolation of PRV strain” to “Isolation of PRV strain” in Figure 4 in revised manuscript according to your suggestion.
Comment 117: L393: split into two sentences: (b-IPMA). Molecular biology approaches include polymerase…
Response: Thank you. We have split into two sentences “(b-IPMA). Molecular biology approaches include polymerase…” in line 393 in revised manuscript according to your suggestion.
Comment 118: l393: quantitative TapMan real-time PCR (qPCR).
Response: Thank you. We have changed “TaqMan real-time (qPCR)” to “quantitative TapMan real-time PCR (qPCR)” in line 393 in revised manuscript according to your suggestion.
Comment 119: L401: widely used in serological approaches, and many PRV…
Response: Thank you. We have changed “widely used approaches of serologic tests, and many PRV” to “widely used in serological approaches, and many PRV…” in line 401 in revised manuscript according to your suggestion.
Comment 120: l403: two sentences: (DIVA). Besides, gB antibodies…
Response: Thank you. We have split into two sentences “(DIVA). Besides, gB antibodies…” in line 403 in revised manuscript according to your suggestion.
Comment 121: l405: can be used for the detection…
Response: Thank you. We have changed “can be used to the detection” to “can be used for the detection…” in line 405 in revised manuscript according to your suggestion.
Comment 122: l415: targeting the gB or gE…
Response: Thank you. We have changed “targeting to the gB or gE” to “targeting the gB or gE…” in line 415 in revised manuscript according to your suggestion.
Comment 123: l416: technology based on the blocking fluorescent…
Response: Thank you. We have changed “technology on the basis of blocking fluorescen” to “technology based on the blocking fluorescent…” in line 416 in revised manuscript according to your suggestion.
Comment 124: l417: less time for PRV detection.
Response: Thank you. We have changed “less time on PRV detection” to “less time for PRV detection” in line 417 in revised manuscript according to your suggestion.
Comment 125: l418: vaccinated pigs whereas a commercial gE-ELISA kit is not.
Response: Thank you. We have changed “vaccinated pigs by comparison with a commercial gE-ELISA kit” to “vaccinated pigs whereas a commercial gE-ELISA kit is not” in line 418 in revised manuscript according to your suggestion.
Comment 126: l422: (99.26%) compared to a commercial …
Response: Thank you. We have changed “(99.26%) relative to a commercial” to “(99.26%) compared to a commercial …” in line 422 in revised manuscript according to your suggestion.
Comment 127: l423: time and cost expenses.
Response: Thank you. We have changed “time and cost of detection” to “time and cost expenses” in line 423 in revised manuscript according to your suggestion.
Comment 128: l426: expected to become new clinical laboratory diagnostic methods for…
Response: Thank you. We have changed “expected to become a new clinical laboratory diagnostic method for” to “expected to become new clinical laboratory diagnostic methods for…” in line 426 in revised manuscript according to your suggestion.
Comment 129: L430: PRV genes including gE, gI…
Response: Thank you. We have changed “PRV genes which contains gE, gI” to “PRV genes including gE, gI…” in line 430 in revised manuscript according to your suggestion.
Comment 130: l432: quantitative TaqMan real-time PCR (qPCR).
Response: Thank you. We have changed “TaqMan real-time (qPCR)” to “quantitative TaqMan real-time PCR (qPCR)” in line 432 in revised manuscript according to your suggestion.
Comment 131: l435: most frequently used approaches…
Response: Thank you. We have changed “most frequently approaches” to “most frequently used approaches…” in line 435 in revised manuscript according to your suggestion.
Comment 132: l444: not suitable for wide range clinical detection…
Response: Thank you. We have changed “not suitable for widely clinical detection” to “not suitable for wide range clinical detection…” in line 444 in revised manuscript according to your suggestion.
Comment 133: L451: to prevent PR. The methods mainly include virus isolation…
Response: Thank you. We have split into two sentences, and the current sentence is “to prevent PR. The methods mainly include virus isolation…” in line 451 in revised manuscript according to your suggestion.
Comment 134: l459: and are time-consuming…
Response: Thank you. We have changed “and time-consuming” to “and are time-consuming…” in line 459 in revised manuscript according to your suggestion.
Comment 135: l460: rapid detection of PRV has been developed.
Response: Thank you. We have changed “rapid detection of PRV is developed” to “rapid detection of PRV has been developed” in line 460 in revised manuscript according to your suggestion.
Comment 136: L469: hundred years. Vaccination…
Response: Thank you. We have split into two sentences, and the current sentence is “hundred years. Vaccination…” in line 469 in revised manuscript according to your suggestion.
Comment 137: l470-474: too long sentence.
Response: Thank you. We have split into two sentences, and the current sentence is “Most PRV vaccines are live gene-modified virus vaccines (Table3).The initial live gene-modified vaccines (attenuated Bartha-K61 strain and PRV Bucharest strain) are usually obtained from extensive passaging of virulent field isolates in cell cultures in 1961, fol-lowing the wide application in pig herds, and effectively controlled PR worldwide” in line 470-474 in revised manuscript.
Comment 138: section 6.1: please explain on which scientific basis some vaccines are based on 1 or 2 or 3 or 4 genes?
Response: Thanks for your question. The scientific basis of PRV 1/2/3/4 genes-deletion in some vaccines as follows: Some genes are important for viral virulence. In particular, it has been confirmed that gE, gI and TK genes are close to the virulence of PRV strains, while they do not affect viral immunogenicity. Therefore, these genes are ideal targets for researchers to create genetic engineering vaccines against PR.
Comment 139: L512: problem of police size.
Response: Thank you. We have corrected the police size of rHN1201TK−/gE−/gI−/11k−/28k− in line 512 in revised manuscript according to your suggestion.
Comment 140: L527 please rephrase.
Response: Thank you. We have changed “Incredibly, the use of Bacillus subtilis as a recombinant vaccine expressing PRV gC and gD proteins can effectively induce mucosal immune response against this disease in recent study” to “Surprisingly, recombinant targeted Bacillus subtilis vaccine expressing PRV gC and gD proteins can effectively induce mucosal immune response against this disease in re-cent study” in line 527 in revised manuscript according to your suggestion.
Comment 141: l539: what DIVA refers to?.
Response: Thanks for your question, and we are apologized for we did not describe it clearly. We have added the full name of DIVA, it refers to “distinction between the infected and vaccinated animals (DIVA)”, and we have added it to the revised manuscript according to your suggestion.
Comment 142: l544: clinical signs, and in adult red fox…
Response: Thank you. We have changed “Incredibly, the use of Bacillus subtilis as a recombinant vaccine expressing PRV gC and gD proteins can effectively induce mucosal immune response against this disease in recent study” to “clinical signs, and in adult red fox…” in line 544 in revised manuscript according to your suggestion.
Comment 143: L557: replace “traditional Chinese medicine” by “this medicine”.
Response: Thank you. We have changed “traditional Chinese medicine” to “this medicine” in line 557 in revised manuscript according to your suggestion.
Comment 144: l562: especially.
Response: Thank you. We have changed “specially” to “especially” in line 562 in revised manuscript according to your suggestion.
Comment 145: l563: based on these bioactivities, it appears that Res…
Response: Thank you. We have changed “Based on these bioactivities, it found that Res” to “based on these bioactivities, it appears that Res…” in line 563 in revised manuscript according to your suggestion.
Comment 146: l571-572: PRV infection, and they need to be further studied for promoting it to be an effective choose for animals…
Response: Thank you. We have changed “PRV infection, and it needs to further study for promoting it to be an effective choose of animals” to “PRV infection, and they need to be further studied for promoting it to be an effective choose for animals…” in line 571-572 in revised manuscript according to your suggestion.
Comment 147: l578: cannot be verified for the inhibition…
Response: Thank you. We have changed “cannot be verified the inhibition” to “cannot be verified for the inhibition…” in line 578 in revised manuscript according to your suggestion.
Comment 148: l578: why they cannot be verified in vivo??
Response: Thanks for your question, and we are apologized for we did not describe it clearly. Some researchers found that compounds had anti-PRV infection activity in vitro, such as kaempferol, panax notoginseng polysaccharides, germacrone, plantago, quercetin, istis indigotica, radix isatidis, marine Bacillus S-12–86 lysozyme, diammonium glycyrrhizin, vanadium-substituted Heteropolytungstate, graphene Oxide, ivermectin and phosphonoformate sodium, but they did not conduct animal experiments to verified if those compounds against PRV infection.
Comment 149: L581: it was found that quercetin can indeed reduce the extent… (nothing obvious here).
Response: Thank you. We have changed “it found that quercetin can obviously reduce the extent” to “it was found that quercetin can indeed reduce the extent…” in line 581 in revised manuscript according to your suggestion.

Round 2
Reviewer 2 Report
no more questions.
It will be better, if the authors can add more key-related references in the part of " PRV entry into the peripheral nervous system (PNS) neurons and spread to the central nervous system (CNS)", such as Radomir Kratchmarov et al., J Virol. 2013; Tal Kramer et al., Cell Host Microbe . 2012; B N Smith et al., Proc Natl Acad Sci U S A . 2000; Fan Jia et al., Front Neuroanat . 2019.